# A phosphorylation-based switch controls TAA1-mediated auxin biosynthesis in plants

Qian Wang [1,2,3,5], Guochen Qin[1,5], Min Cao[1,2], Rong Chen [1], Yuming He[2], Liyuan Yang[2], Zhejun Zeng[2], Yongqiang Yu[2], Yangtao Gu[2], Weiman Xing[1], W. Andy Tao[4] & Tongda Xu [2]*

Auxin determines the developmental fate of plant tissues, and local auxin concentration is precisely controlled. The role of auxin transport in modulating local auxin concentration has been widely studied but the regulation of local auxin biosynthesis is less well understood. Here, we show that TRYPTOPHAN AMINOTRANSFERASE OF ARABIDOPSIS (TAA1), a key enzyme in the auxin biosynthesis pathway in *Arabidopsis thaliana* is phosphorylated at Threonine 101 (T101). T101 phosphorylation status can act as an on/off switch to control TAA1-dependent auxin biosynthesis and is required for proper regulation of root meristem size and root hair development. This phosphosite is evolutionarily conserved suggesting post-translational regulation of auxin biosynthesis may be a general phenomenon. In addition, we show that auxin itself, in part via TRANS-MEMBRANE KINASE 4 (TMK4), can induce T101 phosphorylation of TAA1 suggesting a self-regulatory loop whereby local auxin signalling can suppress biosynthesis. We conclude that phosphorylation-dependent control of TAA1 enzymatic activity may contribute to regulation of auxin concentration in response to endogenous and/or external cues.

[1] Shanghai Center for Plant Stress Biology, Centre for Excellence in Molecular Plant Sciences, Shanghai Institutes for Biological Sciences, Chinese Academy of Sciences, Shanghai 201602, People's Republic of China. [2] FAFU-Joint Centre, Horticulture and Metabolic Biology Centre, Haixia Institute of Science and Technology, Fujian Agriculture and Forestry University, Fuzhou, Fujian 350002, People's Republic of China. [3] University of Chinese Academy Sciences, Beijing 100049, People's Republic of China. [4] Department of Biochemistry, Purdue University, West Lafayette, IN 47907, USA. [5] These authors contributed equally: Qian Wang, Guochen Qin. *email: tdxu@sibs.ac.cn

The dynamics of hormone concentrations are essential for developmental processes in both plants and animals[1,2]. As widely reported in multicellular eukaryotic systems, different concentrations of each hormone can trigger diverse developmental outputs. An imbalance in hormone levels can cause a range of illnesses in animals and abnormal development in plants[3,4]. Auxin is a highly concentration-dependent hormone in plants that regulates all major aspects of plant development. It has been intensively reported that the dynamic auxin concentration gradients are established along with the plant organs, such as roots and that the local concentration of auxin determines distinct developmental fates[4–7]. Furthermore, local auxin concentration adjusts quickly in response to environmental changes[8,9]. Polar auxin transport plays an important role to modulate local auxin levels[10,11], but local auxin production determines the overall auxin content within a plant and also plays an essential role in plant development and in adaptation to environmental changes[7,8].

Indole-3-acetic acid (IAA), the primary natural auxin in plant, is synthesized by multiple pathways[8,12,13]. The indole-3-pyruvic acid (IPA) pathway mediated by TAA1/TARs (Tryptophan Aminotransferase of *Arabidopsis* 1/ Tryptophan Aminotransferase Related proteins) and YUC (YUCCA) is a well-established auxin biosynthesis pathway that contributes the majority of free IAA production[14,15] and is required for major developmental processes, such as embryogenesis, organogenesis, and organ growth[5,8,16]. In *Arabidopsis* root development, the dynamically maintained meristem, elongation, and maturation zones are tightly linked to the local concentration of auxin[5,6]. Accumulation of auxin promotes cell division while lower auxin concentration triggers cell differentiation, which determines root meristem size[17]. Auxin stimulates root hair development in the maturation zone as a way for plants to adapt to environmental changes[18]. As previously reported, auxin biosynthesis mutants show strong defects in both root apical meristem and root hair development[16,19]. Moreover, TAA/YUC-mediated auxin biosynthesis optimizes plant growth in response to a range of environmental changes[6,8,12,18,20]. In these cases, the spatial-temporal regulation of *TAA/YUC* gene transcription modulates auxin biosynthesis. For example, nutrition signals, such as glucose and nitrate induce auxin production by the transcriptional regulation of *YUC2/8/9* and *TAA1/TAR2*, respectively, to regulate root growth[21–24]. Environmental stresses, such as aluminum can modulate root architecture through the control of auxin levels via modulation of *TAA1* gene transcription[25]. Although the transcriptional regulation of these auxin biosynthesis enzymes plays important roles in the control of overall auxin content, non-transcriptional regulation of these enzymes in plants has never been reported. Here, we show a phosphorylation-based mechanism that controls auxin biosynthesis in regulation of plant development. The phosphorylation of an evolutional conserved residue (Threonine 101, T101) on AtTAA1 protein determines its enzymatic activity that further controls auxin biosynthesis. TRANS-MEMBRANE KINASE 4 (TMK4), a kinase in auxin signalling, targets to this phosphorylation site on TAA1 protein, which contributes to the modulation of auxin concentration during plant development.

## Results

**Phosphorylation at T101 regulates AtTAA1 enzymatic activity.** To investigate the underlying regulatory mechanism of auxin biosynthesis at the non-transcriptional level, we used mass spectrometry (MS) to identify the potential protein modifications of auxin biosynthesis enzymes in *Arabidopsis*. First, we analysed TAA1, the key enzyme that converts tryptophan to IPA in the presence of the pyridoxal phosphate (PLP) cofactor[9,16]. We generated *pTAA1-TAA1-GFP* transgenic plants, treated these with a phosphatase inhibitor to prevent protein dephosphorylation, and used immunoprecipitated TAA1-GFP proteins for mass spectrometric analysis (Supplementary Fig. 1). Interestingly, we identified a phosphorylation site at T101 within the in vivo immunoprecipitated TAA1 protein (Fig. 1a). According to the TAA1 protein structure, the T101 residue is located within the PLP binding pocket, indicating that phosphorylation of TAA1 at T101 may affect TAA1 enzymatic activity (Fig. 1b). To verify this, we mutated the T101 residue to aspartic acid (T101D) to mimic the phosphorylation state and tested its enzymatic activity in vitro. We purified various mutated TAA1 proteins from *Escherichia coli* and separated the proteins using a native gel, then stained the gel using a catalytic reaction buffer (Method section). In this way, the active TAA1 would catalyse transamination reaction then result in a dark colour in the gel[16,26]. TAA1[K217A] protein was set as a control, as the K217 residue is reported to be required for PLP binding[16]. In contrast to the TAA1[WT] protein, which displayed the colour of reaction products in the gel, TAA1[T101D] protein was not active in the assay suggesting that *E. coli*-purified TAA1[T101D] proteins completely lost their transaminase catalytic activity similar to the TAA1[K217A] proteins (Fig. 1c). We used another liquid assay[27] and confirmed that the TAA1[T101D] proteins were unable to convert tryptophan to IPA (Fig. 1d). To investigate whether the non-phosphorylated form has the opposite effect on TAA1 enzymatic activity to the phosphorylated form, we mutated T101 to alanine (T101A). Interestingly, we found TAA1[T101A] proteins exhibited similar or even higher enzymatic activity compared with TAA1[WT] only when PLP cofactors were present in a sufficient amount in the assay (Fig. 1c, d; Supplementary Fig. 2a, b). Under PLP-deficient conditions, neither TAA1[T101A] or TAA1[T101D] proteins showed normal enzymatic activity due to an abnormal ability to capture PLP from *E. coli* which is distinct from *E. coli*-purified TAA1[WT] proteins that are already bound to PLP[16] (Supplementary Fig. 2a–c). This indicates that the T101 residue is indeed important for PLP-dependent enzymatic activity of TAA1 and that the T101A mutation does not fully simulate non-phosphorylated T101.

To further investigate the function of T101 phosphorylation in vivo, we introduced the TAA1 promoter-driven *TAA1[WT]*, *TAA1[T101D]*, and *TAA1[T101A]* into the *wei8-3* (*TAA1* is also known as *weak ethylene insensitive 8, WEI8*) mutant which is a strong TAA1 loss-of-function mutant[16]. Consistent with previous work[16,19], blocking TAA1 function in *Arabidopsis* by either genetic mutation or the chemical inhibitor L-kynurenine (L-Kyn)[28] impairs the root apical meristem and root hair development, and this can be rescued by exogenous auxin application, thus providing a good system to study how auxin levels are controlled (Supplementary Fig. 3). Compared with *TAA1[WT]*, we found that *TAA1[T101D]* could not complement either the root meristem or the root hair phenotype in the *wei8-3* mutant, indicating an abolished function of TAA1[T101D] in vivo (Fig. 1e, f; Supplementary Fig. 4). *TAA1[T101A]* only partially rescued the mutant phenotype and could not fully rescue root meristem phenotype of *wei8-3;tar2-1* mutant, suggesting that TAA1[T101A] is not fully functional in vivo even though PLP slightly increased root meristem size of *TAA1[T101A]; wei8-3* (Fig. 1e, f; Supplementary Fig. 4; Supplementary Fig. 5). These results are consistent with the biochemical results and indicate that TAA1 activity is switched off when T101 is in a phosphorylated state in plants.

**Dominant effect of T101D mutation on TAA1 proteins.** In *Arabidopsis*, the cytosolic-localized TAA1, TAR1, and the

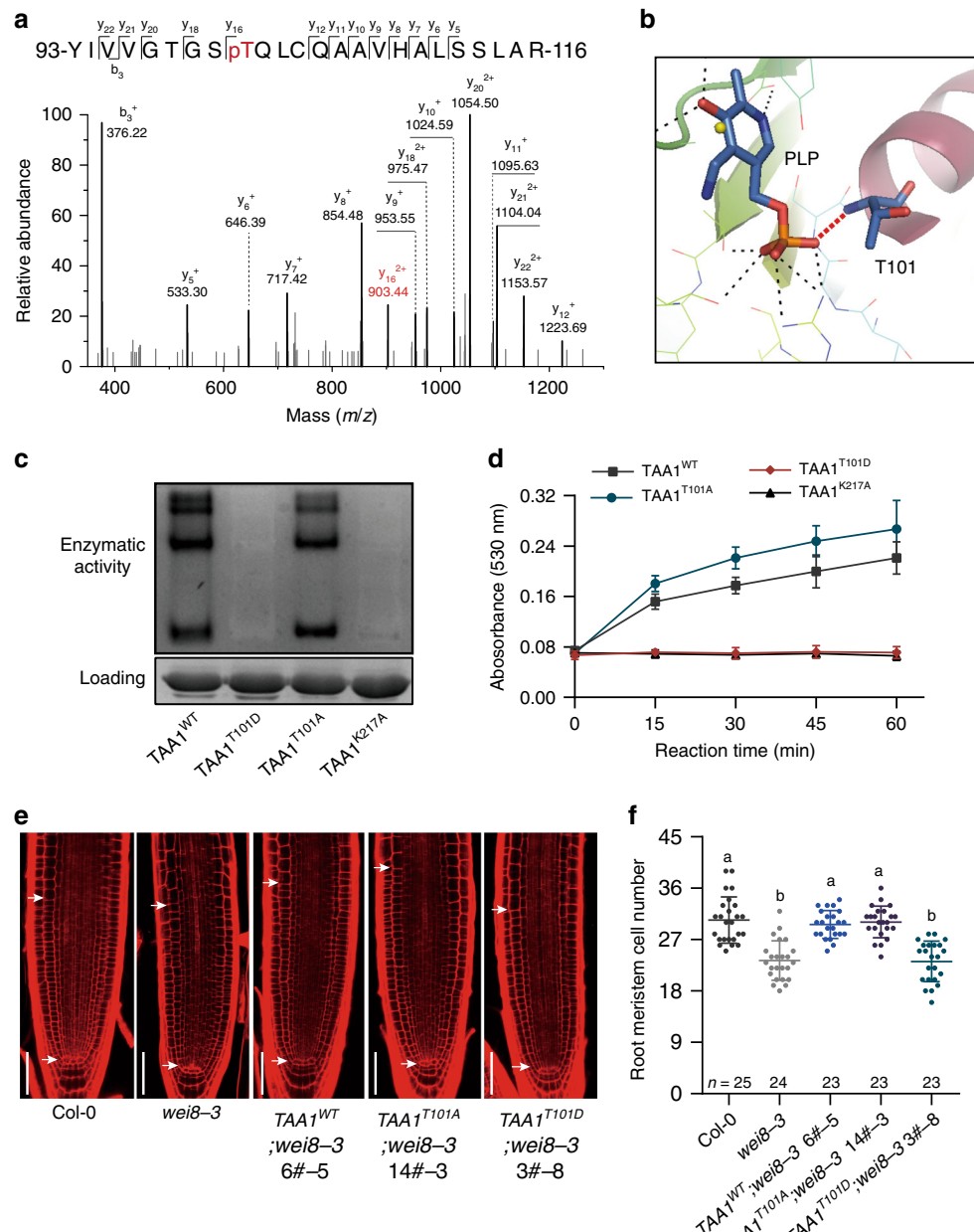

**Fig. 1 Phosphorylation modification on T101 residue turns off TAA1 enzymatic activity. a** Mass spectrometric analysis showing the phosphorylation modification at Thr101 site on TAA1. **b** Structural simulation of the connection between T101 residue and PLP cofactor. **c** In gel tryptophan aminotransferase activity (Trp-AT) of *E. coli*-purified GST-TAA1[WT], GST-TAA1[T101A], and GST-TAA1[T101D]. GST-TAA1[K217A] was set as a control. Protein amount shown using Coomassie blue staining. *E. coli*-purified proteins were separated in a native gel followed by the catalytic reaction and further staining. The intensity of dyed bands in the gel represent catalytic activity. Lack of a dark band in the GST-TAA1[T101D] and GST-TAA1[K217A] lanes indicate complete loss of enzymatic activity. Three independent biological repeats showed similar results. **d** Transaminase catalytic activity detected by measuring IPA production with Salkowski reagent (Methods section). Transaminase catalytic reactions were performed in the reaction reagent containing 2.5 μg recombinant protein with a His tag. The products were measured by reading the absorbance at 530 nm at different reaction time points. Values denoted the mean ± s.d. (*n* = three independent biological repeats). **e** Representative pictures of root apical meristem from 5-day-old seedlings in Col-0, *wei8-3*, and *pTAA1-gTAA1* (WT, T101D, T101A);*wei8-3* complementation transgenic plants. White arrows show the meristem zone; Scale bar 50 μm. **f** Quantification of root meristem size in **e**. Three independent lines of *TAA1[WT];wei8-3* and *TAA1[T101D];wei8-3*, and 2/7 lines of *TAA1[T101A];wei8-3* showed similar results. *n* denotes the number of independent seedlings; one-way ANOVA with Tukey multiple comparisons test. Different letters represent significant difference between each other, *P* < 0.0001.

endoplasmic reticulum-localized TAR2 have redundant functions in auxin biosynthesis[9,29]. Interestingly, when over-expressing *TAA1[T101D]* in *wei8-3;tar2-1* mutants, we observed dominant negative effects on plant growth. *35S-TAA1[T101D]; wei8-3;tar2-1* seedlings exhibited a more enhanced phenotype compared to *wei8-3;tar2-1* (Fig. 2a–h). Fifty four out of 342

*35S-TAA1[T101D];wei8-3;tar2-1* seedlings (15.79%) showed a strong phenotype with the abnormal cotyledon and/or the root retardation similar to *wei8;tar1;tar2* triple mutant, while only 8 of 451 (1.77%) of *wei8-3;tar2-1* seedlings showed this phenotype. This indicated that the function of TAR1 in *35S-TAA1[T101D];wei8-3;tar2-1* might be partially abolished.

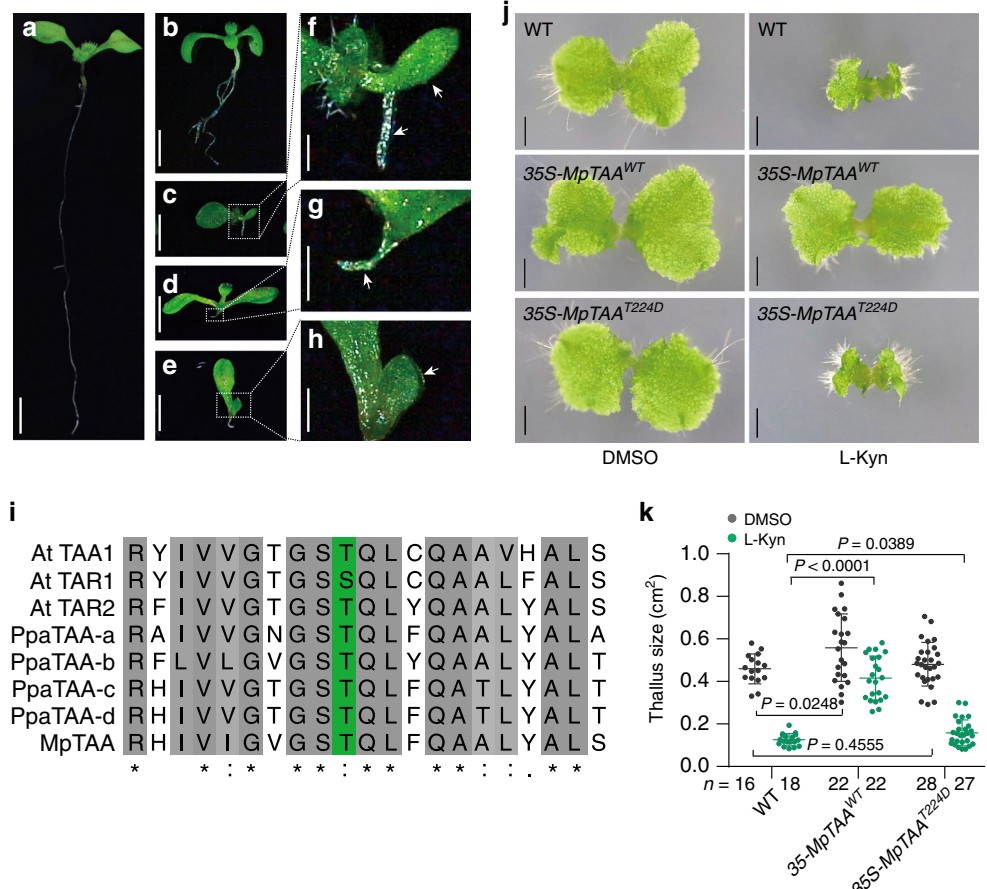

**Fig. 2 The function of T101 phosphorylation on AtTAA1 homologs and in TAA1 orthologs during evolution. a–h** Representative images of 7-day-old seedling in Col-0 **a**, *wei8-3;tar2-1* **b**, *wei8-3;tar2-2;tar1-1* segregated from *wei8-3;tar2-2* crossed with *wei8-3;tar1-1* F2 generation **c**, and *35S-TAA1^T101D^;wei8-3; tar2-1* **d**, **e**. Magnified picture of *wei8-3;tar2-2;tar1-1* **f** and *35S-TAA1^T101D^;wei8-3;tar2-1* **g**, **h**. 54 out of 342 *35S-TAA1^T101D^;wei8-3;tar2-1* seedlings (15.79%) from three independent lines (T2 generation) showed strong phenotype. Source data were provided in the source file. The result represents one of two replicates which showed a similar phenomenon. Arrows show asymmetric cotyledons and severe root defects. Scale bar, 0.25 cm **a–e**; 40 mm **f–h**. **i** Sequence alignment of TAA orthologs among *Arabidopsis* (*At*), *Physcomitrella Patterns* (*Ppa*), and *Marchantia Polymorpha* (*Mp*). Green colour column indicates the conserved residues as the AtTAA1 T101 site. **j** Phenotype of overexpressing *MpTAA^WT^* and *MpTAA^T101D^* in wild-type *Marchantia* (Tak-1) treated with 500 μM L-Kyn. Images denote representative thalli. Scale bar, 0.2 cm. **k** Quantification of the thallus size in **j**. Gammae from three independent lines of *35-MpTAA^WT^* and two independent lines of *35-MpTAA^T224D^* at 16 day old were quantified by thalli area. *n* denotes the number of independent thallus; two-sided *t*-test. *P* values were shown as indicated.

Interestingly, we found that TAA1 and TAR1 were able to form heterodimers as tested by coimmunoprecipitation assay (Supplementary Fig. 6). Together, these data suggest that phosphorylation of the TAA1 protein at the T101 site not only controls the on–off switch of TAA1 enzymatic activity but may also affect the functions of other TAA1 homologs by dimerization.

**T101 phosphorylation on TAA1 is evolutionarily conserved.**
When we examined the TAA orthologs in other plant species, we discovered that the T101 residue is remarkably conserved across different plant species (Fig. 2i; Supplementary Fig. 7), including evolutionary distant species, such as *Marchantia Polymorpha*, in which auxin signalling is already present despite being ancestral to land plants[30–32]. This finding implies that the T101-mediated switch of AtTAA1 may be conserved during evolution. To provide evidence for this hypothesis, we tested whether MpTAA was also phosphorylated by MS using *35S-MpTAA* transgenic *Marchantia*. We found Thr224 on MpTAA protein, a conserved residue analogous to T101 of AtTAA1, was also phosphorylated in vivo (Fig. 2i; Supplementary Fig. 8). Then we tested the

function of a phosphor-mimic mutation of MpTAA in transgenic *Marchantia*. As expected, the phosphor-mimic version (T224D) of MpTAA also affected the function of MpTAA (Fig. 2j, k; Supplementary Fig. 8a, b). It has been reported that MpTAA mediates auxin biosynthesis and is essential for the development of *Marchantia*[33–35]. Both the mutation of MpTAA and the treatment with the TAA inhibitor (L-Kyn) suppress thallus growth, while overexpression of MpTAA in wild-type *Marchantia* (Tak-1) can enhance its resistance to the L-Kyn treatment[33]. *35S-MpTAA^WT^* transgenic *Marchantia* grew much better than wild-type (Tak-1) thallus on medium containing a high concentration of L-Kyn but *35S-MpTAA^T224D^* showed strong growth inhibition even though the average thallus size was slightly bigger than in wild-type plant (Fig. 2j, k). These results suggest that the phosphorylation status of T224 on the *MpTAA* protein also controls its function in *Marchantia*, and that this phosphorylation-dependent mechanism is well conserved throughout plant evolution.

**TAA1 protein is phosphorylated by TMK4.** After identifying the phosphorylation modification that regulates TAA1, we speculated

that TAA1 could be targeted and phosphorylated by protein kinases. To identify the potential kinases that regulate TAA1, we performed a mass spectrometric analysis of the TAA1 complex proteins in young *Arabidopsis* seedlings and found candidate protein kinases. We then used the unbiased yeast two-hybrid assay to screen the interactions between protein kinase candidates and TAA1. We found that TMK4, a member of the transmembrane kinase family proteins in the regulation of auxin signalling[36–38], specifically interacted with TAA1 in yeast (Fig. 3a). We then confirmed this interaction using an in vitro pull-down assay (Fig. 3b). In addition, we found that TMK4 was coimmunoprecipitated with TAA1 in vivo (Fig. 3c). Thus, although TMK4

proteins localized at the plasma membrane and TAA1 proteins distributed in the cytosol, which we observed in transgenic plants driven by native promoters (Supplementary Fig. 9), a portion of cytosolic TAA1 at the submembrane was targeted by TMK4.

Using an in vitro kinase assay, we showed that TMK4 phosphorylated TAA1 directly (Fig. 3d). To verify the phosphorylation sites targeted by TMK4, we detected the phosphor-peptide of TAA1 by MS after the kinase reaction in vitro, and found the pT101-containing phosphor-peptide indicating TMK4 could phosphorylate TAA1 at the T101 site (Supplementary Fig. 10a). By mass spectrometric analysis, we further quantified the phosphorylation level of T101 on TAA1 protein in vivo and

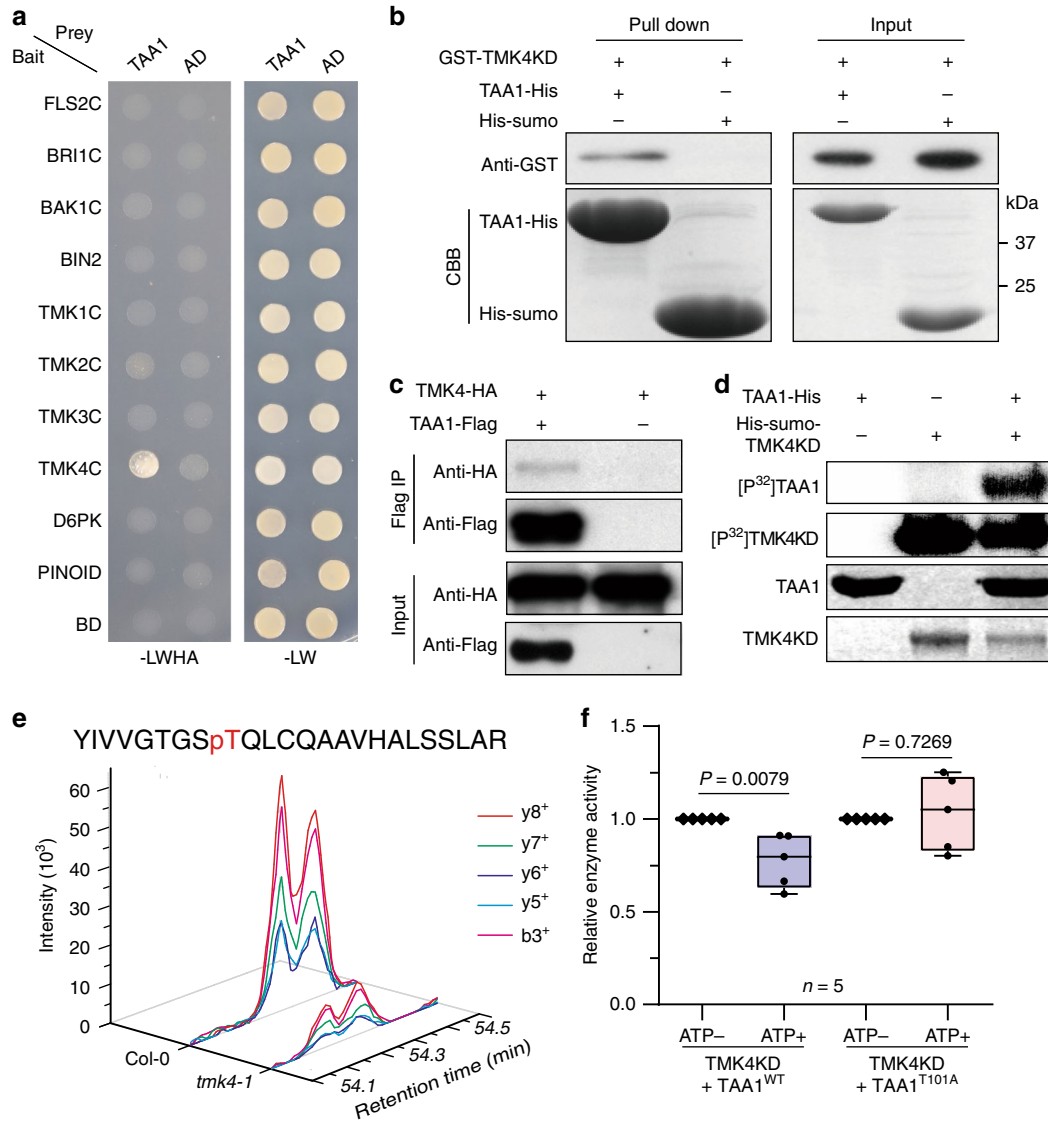

**Fig. 3 T101 of TAA1 is targeted by TMK4 kinase. a** Yeast two-hybrid assay of multiple protein kinases (bait) and TAA1 protein (prey). Three independent biological repeats. **b** In vitro pull-down assay showing GST-TMK4 kinase domain (KD) can be pulled down by TAA1-His proteins. His-Sumo proteins were used as a negative control. TMK4 KD was detected via western blot using an anti-GST antibody. TAA1-His and His-Sumo shown by Coomassie blue staining (CBB). Three independent repeats showed similar results. **c** TMK4 was coimmunoprecipitated with TAA1 in protoplasts. 35S-TMK4-HA and 35S-TAA1-Flag were transiently coexpressed in *Arabidopsis* protoplasts. Single transformation of 35S-TMK4-HA was used as control. Three independent repeats showed similar results. **d** In vitro kinase assay of TMK4 kinase domain (KD) on TAA1 protein. Three biological repeats. **e** Mass spectrometric analysis of the phosphor-peptide intensity in Col-0 and *tmk4-1*. The intensity indicates the phosphorylation level of T101 containing phosphor-peptide (93aa–116aa). Standardization and another two repeats were shown in Supplementary Fig. 10b, c. **f** Quantification of TAA1 enzymatic activity with or without TMK4KD phosphorylation. *E. coli*-purified His-sumo-TMK4KD and TAA1 proteins were mixed together to perform a phosphorylation reaction triggered by ATP before the in-gel transaminase catalytic reaction (Methods section). ATP+ indicated the phosphorylation reaction between TMK4KD and TAA1 with ATP, and ATP− indicated the abolished phosphorylation reaction without ATP. *P* values were shown as indicated (*n* represents five biological repeats); two-sided *t*-test. Centre line in box represents mean and whiskers show minimum to maximum.

found that it was decreased in the *tmk4-1* mutant compared to wild type (Fig. 3e; Supplementary Fig. 10b, c). Thus, TMK4 is one of the protein kinase candidates that targets the T101 site on the TAA1 protein.

Considering the phosphorylation status of T101 was able to determine the on–off switch of TAA1 enzymatic activity, we further verified whether TMK4 could regulate TAA1 enzymatic activity through phosphorylation at T101. We mixed both TMK4 and TAA1 proteins to perform a phosphorylation reaction triggered by ATP prior the in-gel transaminase catalytic reaction. As expected, we found that TMK4 significantly inhibited the catalytic activity of TAA1$^{WT}$ but not TAA1$^{T101A}$ (Fig. 3f and Methods section). The transaminase activity of total protein from the *tmk4-1* mutant increased significantly compared to wild type, but was rescued in the *tmk4-1;wei8-3* double mutant (Supplementary Fig. 11a), indicating TMK4 negatively modulates TAA1 enzymatic activity in vivo. Meanwhile, we found that TMK4 did not obviously affect the *TAA1* gene transcription and/or TAA1 proteins subcellular localization (Supplementary Fig. 11b, c). Therefore, we have comprehensive evidence that TMK4 interacts and phosphorylates TAA1 at the T101 site to switch off the TAA1 enzymatic activity.

**TMK4 negatively regulates auxin biosynthesis through TAA1**. To confirm whether TMK4 is involved in regulating TAA1-dependent auxin biosynthesis, we tested the *tmk4-1* mutant phenotype during root development. The *tmk4-1* mutant showed an enlarged root meristem and longer root hairs (Supplementary Fig. 12a–f), similar to the phenotype of seedlings treated with auxin. This phenotype was confirmed by another allele, *tmk4-2*, and was complemented by genomic *TMK4* (Supplementary Fig. 12a–f). To further confirm the function of TMK4 in auxin biosynthesis, we directly measured levels of free IAA in the *tmk4-1* mutant. Indeed, we found an increased IAA abundance in the *tmk4-1* mutant that could be rescued by genomic *TMK4* (Supplementary Fig. 12g). In addition, both *DR5-GUS* and DII-Venus auxin markers showed that auxin concentration in *tmk4* mutants was dramatically increased (Fig. 4a, b; Supplementary Fig. 13a, b), consistent with the measurement of free IAA levels.

By introducing the TAA1 mutation to the *tmk4* mutant or blocking TAA1 enzymatic activity by L-Kyn chemical in the *tmk4* mutant, we found that the increased auxin concentration in *tmk4* was restored when using *DR5-GUS* or DII-Venus as an auxin marker (Fig. 4a, b; Supplementary Fig. 13a, b). The increased levels of free IAA in *tmk4-1* was also mostly restored with the introduction of TAA1 mutations (Fig. 4c). Similarly, both the root meristem and the root hair phenotype in the *tmk4-1* mutant were mostly blocked by either the TAA1 mutation or L-Kyn treatment (Fig. 4d, e; Supplementary Fig. 13c). These genetic data strongly indicate that TAA1-based auxin biosynthesis contributes most of the overproduced auxin in *tmk4* and corroborate the biochemical data to support the negative regulation of auxin biosynthesis by TMK4 through TAA1.

TMK4 belongs to a transmembrane kinase family which is a key regulator in auxin signalling[37]. Interestingly, we found that the phosphorylation level of TMK4 proteins increased in the auxin overproduction mutant *yuc1-D*[39] as observed using a gel shift assay, indicating that auxin may activate TMK4 (Supplementary Fig. 14a). It has been reported that auxin controls its own biosynthesis to balance its local concentration[40,41]. When we treated the *Arabidopsis* protoplast with exogenous auxin, we also found an increase in phosphorylation on TAA1 proteins at T101 residue, suggesting a negative feedback regulation of auxin biosynthesis (Supplementary Fig. 14b). The auxin effect on the T101 phosphorylation level of TAA1 proteins was partially

blocked in the *tmk4* mutant (Supplementary Fig. 14b), suggesting the TMK4-TAA1 regulation module might be one of the self-regulation mechanisms to control auxin biosynthesis in plants.

**Discussion**

In plants, TAA/YUC-mediated auxin biosynthesis pathway is a well-established pathway to produce the majority of free IAA[8,14,15]. In depth studies previously revealed that local auxin biosynthesis plays an indispensable role for plants in response to both developmental and environmental cues, and suggesting that accurate regulation of auxin biosynthesis at multiple levels is required[6,8,12,18]. The spatial-temporal regulation of auxin at the transcriptional level has been described as an important mechanism for auxin biosynthesis regulation, yet little is known about the non-transcriptional regulation of this pathway. Here, we establish a phosphorylation-based regulatory mechanism that determines TAA1 enzymatic activity which further controls auxin biosynthesis. To our knowledge, this is the first identified post-translational regulatory mechanism of auxin biosynthesis in plants.

In this work, we identified an in vivo phosphorylation site at T101 residue on TAA1 protein by mass spectrometric analysis and provided further evidence of the mechanistic switch of TAA1 enzymatic activities using several experimental approaches. This regulatory mechanism of auxin biosynthesis is essential for maintaining the well-organized root development in *Arabidopsis* (Fig. 1; Supplementary Fig. 4). Importantly, the T101 phosphorylation site is evolutionarily conserved and functions in the ancient plant *Marchantia*, indicating the significance of this novel regulatory mechanism on auxin biosynthesis (Fig. 2i–k; Supplementary Fig. 8c). The T101 phosphorylation site is located in the PLP binding pocket, which is required for TAA1 enzymatic activity. Furthermore, using functional analysis of both phosphor-mimic TAA1$^{T101D}$ and non-phosphor TAA1$^{T101A}$ mutations, we found that the phosphorylation at the T101 residue might obstruct its binding to the PLP cofactor. Both mutations affected TAA1 binding with the coenzyme PLP, yet TAA1$^{T101A}$ proteins had restored enzymatic activity when supplied with sufficient PLP while the TAA1$^{T101D}$ mutant could not restore enzymatic activity despite the presence of sufficient PLP levels (Fig. 1c, d; Supplementary Fig. 2), suggesting the phosphorylation is one of the key regulatory mechanisms of the TAA1 enzyme. This work also initiates the future directions to identify other regulatory mechanisms, including other phosphorylation sites of TAA1 that regulate auxin concentration in plants.

Next, we identified TMK4, a protein kinase that targets TAA1 protein at the T101 site, as a negative regulator of auxin biosynthesis (Fig. 3; Fig. 4; Supplementary Fig. 10). TMKs have been implicated as key regulators in auxin signalling[37,38]. Here, we found that TMK4 partially participates in auxin regulation of T101 phosphorylation on TAA1 proteins (Supplementary Fig. 14), suggesting that auxin may facilitate a feedback regulation of its biosynthesis that balances the local auxin concentration that is essential for distinctive developmental outputs.

Interestingly, our data indicated that other kinases may also be involved in regulating the TAA1 phosphorylation since TAA1 was still slightly phosphorylated in the *tmk4* mutant (Fig. 3e). It would be interesting to identify additional kinases that can target TAA1 or its homologs. Identifying such kinases would provide evidence for a more comprehensive signalling network in control of auxin biosynthesis for other developmental or environmental responses. We also found that free IAA level and root phenotype in *tmk4* was not fully suppressed by *wei8-3* single mutant (Fig. 4c; Supplementary Fig. 13c). This indicated that, besides TAA1, TMK4 must have other targets involved in auxin

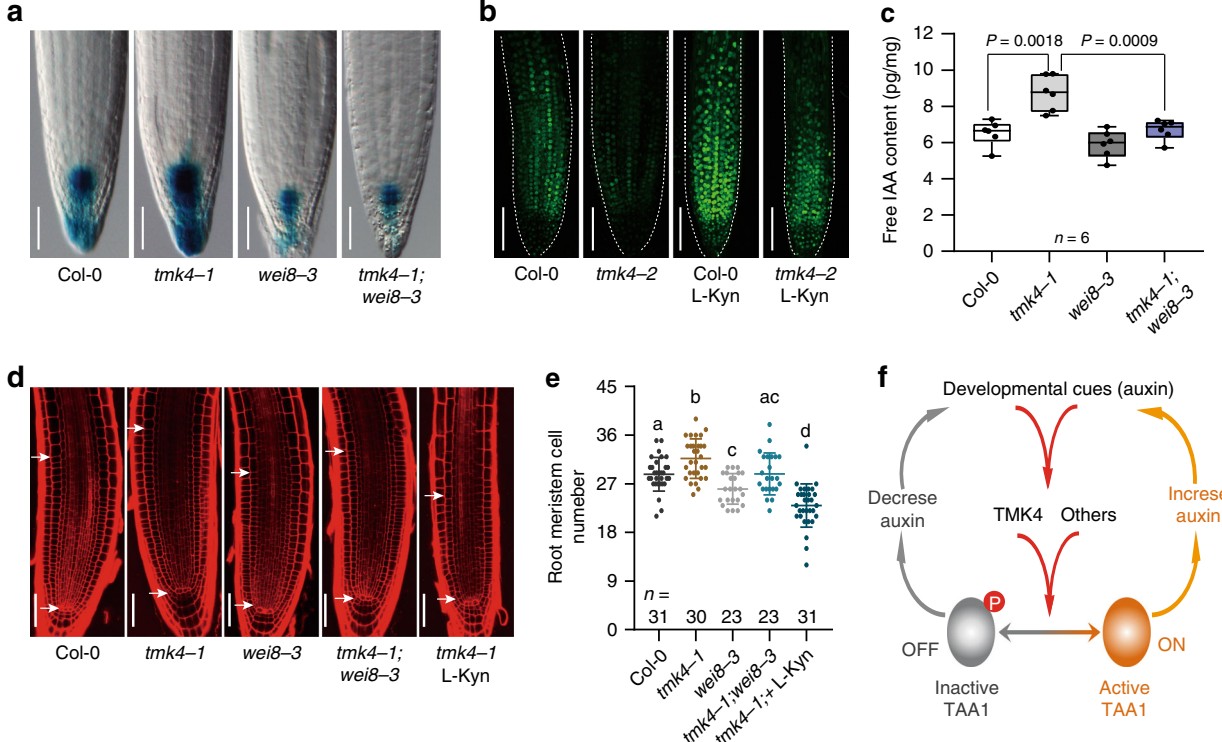

**Fig. 4 TMK4 negatively modulates local auxin biosynthesis through TAA1. a** GUS staining of 5-day-old Col-0, *tmk4-1*, *wei8-3*, and *tmk4-1;wei8-3* crossed with *DR5-GUS* marker. Scale bar, 50 μm. Quantification data and statistical analysis were shown in Supplementary Fig. 13a. **b** DII-Venus signal in the roots of Col-0 and *tmk4-2*. Seedlings were grown on medium with DMSO or 1.5 μM L-Kyn for 5 days. Scale bar, 75 μm. Quantification data and statistical analysis were shown in Supplementary Fig. 13b. **c** Free IAA quantification of Col-0, *tmk4-1*, *wei8-3*, and *tmk4-1;wei8-3*. Six individual biological repeats (*n*) were performed using 8-day-old seedlings. *P* values were shown as indicated; two-sided paired *t*-test. Centre line in box represents mean and whiskers show minimum to maximum. **d** Representative pictures of root apical meristem from 5-day-old seedlings. White arrows show meristem zone. Scale bar, 50 μm. **e** Quantification of root meristem size in **d**. *n* denotes the number of independent seedlings; one-way ANOVA with Tukey multiple comparisons test. Different letters represent significant difference between each other, *P* < 0.05. **f** Working model showing the phosphorylation-dependent mechanism of the on–off switch of TAA1 in plants. This central mechanism may regulate auxin levels in response to various upstream cues, which further modulates plant development and/or helps to deal with environmental changes.

concentration regulation. It would be also interesting to know how auxin signalling modulates auxin biosynthesis through multiple targets.

Altogether, our work illustrates a phosphorylation-dependent switch of TAA1 enzymatic activity that is essential for the regulation of auxin biosynthesis in plants (Fig. 4f). Our observations offer an option for a mechanism of how local auxin biosynthesis is tightly controlled and subsequently regulates the dynamic auxin concentration gradients along with plant tissues under certain developmental or environmental stimulus. Our findings also provide a new understanding on how plants cope with their comprehensive developmental processes, in part, by modulating auxin biosynthesis.

## Methods

**Plant materials and growth conditions**. All plant materials were grown in plant growth chambers (PERVICAL AR-66L3) using long-day condition (16 h light/8 h dark) at 23 °C, unless indicated otherwise. For *Arabidopsis*, Col-0 was used as wild-type plant. *tmk4-1* (GABI_348E01) and *tmk4-2* (SAIL_597_G10) in Col-0 background were used for phenotypic analyses. *wei8-3* (SALK_127890)[16], *tar2-1* (SALK_021258), and *tar2-2* (SALK_137800) were used for genetic crossing and TAA1 function analyses. To generate *pTMK4-TMK4-GFP* transgenic plants, 3.3 kb *TMK4* promoter and *TMK4* genomic coding region were amplified by polymerase chain reaction (PCR) and inserted into a pDONR-Zeo vector, subsequently transferred into a pGWB504 destination vector by LR reaction and then transformed into *tmk4-1*. To generate *pTAA1-TAA1^WT-GFP* transgenic plants, 2 kb promoter region of *TAA1* and *TAA1* genomic coding region were amplified by PCR and inserted into a pDONR-Zeo vector, subsequently transferred into a pGWB504 destination vector and then transformed into Col-0 and *wei8-3* mutants.

To generate *35S-TAA1^WT-GFP* transgenic plants, *TAA1* CDS (coding sequence) was amplified by PCR and inserted into a pDONR-Zeo vector, subsequently transferred into a pGWB505 vector and then transformed into *wei8-3;tar2-1*. *pTAA1-TAA1^{T101D/A}-GFP* and *35S-TAA1^{T101D}-GFP* constructs were generated by site-directed mutagenesis using KOD plus DNA polymerase (Toyobo KOD-201), and then transformed into *wei8-3* or *wei8-3;tar2-1*. Site mutation primers were shown in Supplementary Table 2.

For *M. polymorpha*, strain Takaragaike-1 (Tak-1) was used as wild type[42]. *MpTAA* CDS was amplified by PCR and inserted into entry vector pDONR-Zeo. pDONR-MpTAA^{T224D} was generated by site-directed mutagenesis and transferred into a pGWB511 destination vector by LR reaction, and then transformed into Tak-1 to generate *35S-MpTAA^{WT}-Flag* and *35S-MpTAA^{T224D}-Flag* transgenic plants.

**Root meristem and root hair phenotypic analyses**. Five-day-old seedlings with similar growth condition were used for root meristem phenotypic analysis after stained with 50 μg/mL PI (Propidiumiodide, Sigma Cat# P4170) for 1 min. Images of root meristem were taken by Leica SP8 confocal microscope (Leica TCS SP8 X; excitation: 514 nm, emission: 600–650 nm). The number of cortex cells in a file extending from the quiescent centre to the first elongated cortex cell in root was defined as root meristem size and displayed as root meristem cell number[17,43]. Root hair images were taken by Leica M205FA Microscope or SMZ-18 stereomicroscope (Nikon SMZ18). The fixed region (1 mm length) 2 mm far away from the root tip was selected for root hair length measurement using ImageJ software (v1.8.0).

**Identification of phosphorylation sites in vivo by LCMS/MS**. Seven-day-old *pTAA1-TAA1-GFP* transgenic *Arabidopsis* seedlings grown in long-day conditions were collected and treated with 50 μM cantharidin (Sigma C7632) for 6 h. Total proteins were extracted with extraction buffer (50 mM Tris-HCl (pH 7.5), 150 mM NaCl, 5 mM EDTA, 5% glycerol, 1% TritonX-100 with protease inhibitor, phosphatase inhibitor, and PMSF). GFP-Trap agarose beads (GFP-Trap®_A, gta-20,

ChromoTek) were used to immunoprecipitate TAA1-GFP proteins. The proteins were separated on a sodium dodecyl sulfate–polyacrylamide gel electrophoresis (SDS–PAGE) gel and cut according to molecular weight after Coomassie staining, and followed by liquid chromatography with tandem MS (LCMS)/MS analysis. For comparing T101 phosphorylation levels in Col-0 and tmk4-1, pTAA1-TAA1;Col-0, and pTAA1-TAA1;tmk4-1 transgenic plants were used for immunoprecipitating TAA1 proteins. The amount of TAA1-GFP proteins was standardized by two non-phosphorylated peptides. To detect auxin effects on T101 phosphorylation level, 35S-TAA1-GFP plasmids were transformed into Col-0 and tmk4-1 protoplasts incubated for 8 h. The protoplasts were collected after treated with EtOH (as mock) or 500 nM IAA for 10 mins. TAA1-GFP proteins were immunoprecipitated from the protoplasts further used for quantitative mass spectrometric analysis.

Protein peptides were reconstituted in 4 μL of 0.1% formic acid (FA) and then were injected onto a NanoAcquity UPLC system (Waters, USA) equipped with a C18 capillary trapping column (20 mm × 180 μm, Waters PN:186006527) and a C18 analytical column (75 μm × 150 mm, Waters PN:186003543), and separated over a 70-min gradient at 350 nL/min. This HPLC system was online with an Orbitrap Fusion MS (Thermo Fisher, USA) using high-energy collision-induced dissociation mode. MS survey scan was performed at a resolution of 120,000 over the m/z range of 300–1800, and MS/MS was selected by data-dependent scanning on the ten most intense ions (automatic gain control (AGC) 1E5, maximum injection time 100 ms). Dynamic exclusion was set to a period of 60 s.

MS/MS data were searched against the protein sequences of TAA1 using Mascot Daemon 2.5 (Matrix Science UK). The search parameters were restricted to tryptic peptides at a maximum of two missed cleavages. Cysteine carbamidomethylation was designated as a fixed modification. Oxidation of methionine and serine, threonine or tyrosine phosphorylation were considered as variable modifications. Mass tolerances were set up to 10 ppm for MS ions and 0.8 Da for MS/MS fragment ions. Peptide assignments were filtered by an ion score cutoff of 15.

To monitor phosphorylation changes on TAA1 proteins, the parallel reaction monitoring (PRM) was applied for phosphor-peptides quantification. Peptide samples were dissolved in 6 μL 0.1% FA and injected 4 μL into an EASY-nLC 1200 HPLC system (Thermo Fisher, USA). Eluent was introduced into the mass spectrometer, using an in-house C18 capillary column packed with 2.2 μm C18 particles (Polymicro; 75 μm inner diameter and 12 cm of bed length). The mobile phase buffer consists of 0.1% FA in water (buffer A) with an eluting buffer of 0.1% FA in 80% (v/v) acetonitrile (buffer B). The gradient was set as 5–25% buffer B for 40 mins and 25–45% for 10 mins at 300 nL/min. The sample was acquired on an Orbitrap Fusion MS (Thermo Fisher, USA). Each sample was analysed under PRM with an isolation width of ±1.6 Da. In all experiments, a full mass spectrum at 120,000 resolutions (AGC target 2E5, 50 ms maximum injection time) was followed by PRM scans at 30,000 resolutions (AGC target 1E5, 200 ms maximum injection time). PRM data were manually curated within Skyline (version 4.1.0.18163)[44].

Similar experiment was carried out for detection of MpTAA phosphorylation sites. Three-week-old 35S-MpTAA transgenic thalli grown on 1/2 B5 medium were collected for protein extraction (50 mM Tris-HCl (pH 7.5), 150 mM NaCl, 5 mM EDTA, 5% glycerol, 1% TritonX-100 with protease inhibitor, phosphatase inhibitor, and PMSF). MpTAA proteins were isolated by anti-Flag beads (Sigma cat. #A2220) then separated on 8% SDS–PAGE gel followed by Coomassie staining. The proteins were cut according to molecular weight then send for mass spectrometric analysis.

**Free IAA quantification.** The IAA chemicals in plants were extracted by the published method[9,16] with slight modifications. Briefly, ~50 mg fresh seedlings were collected in a 1.5 mL tube and then homogenized in liquid nitrogen using a mill at 60 Hz for 1 min. A total of 1 mL cold sodium phosphate buffer (50 mM, pH 7.0) containing 0.02% sodium diethyldithiocarbamate and 1 ng [$^{13}C_6$]-IAA (CLM-1896-0.01, Cambridge Isotope Laboratories, USA) as an internal standard were added into the tube, which was shaken at 4 °C for 25 min and then centrifuged at 20,000×g for 15 mins. The supernatant was transferred into a new 1.5 mL tube and adjusted to pH 2.7 by adding 1 M hydrochloric acid. IAA in extraction solution was purified by solid-phase extraction (SPE) using a Waters C18-SPE column (1cc/100 mg; WAT023590) that was conditioned with 3 mL ACN (0.1% FA) and equilibrated with 3 mL H$_2$O (0.1% FA) before use. After loading the sample, the SPE column was washed with 1 mL 5% ACN (0.1% FA). IAA was eluted with 0.5 mL 75% ACN (0.1% FA) and 50 μL eluate was analysed by an Acquity UPLC system (Waters, USA) on a BEH C18 column (2.1 × 150 mm, 1.7 μm, Waters PN:186002353) with an online MS detection using TripleTOF 5600 + (AB Sciex, USA) in a positive electrospray mode. IAA concentrations were determined by comparing IAA peak areas with those of [$^{13}C_6$]-IAA. IAA contents (pg IAA per mg of fresh seedlings) were calculated for statistical analysis.

**In-gel tryptophan aminotransferase activity assay.** The in-gel tryptophan aminotransferase (Trp-AT) activity assay was performed as previous reported[16,45]. Briefly, TAA1$^{WT/T101A/T101D}$ CDS were amplified and inserted into PGEX4T-2 vector. GST-TAA1 was purified from E. coli, separated by a native PAGE gel under non-denaturing condition at 4 °C. The gel was washed by precooled ddH$_2$O and incubated with a reaction buffer (0.1 M Tris-HCl (pH 8.6), 5.5 mM

tryptophan, 0.2 mM PLP, 12.5 mM α-ketoglutaric acid, 98 μM phenazine methosulfate, and 0.6 mM nitroblue tetrazolium) at 37 °C in the dark until the appearance of bands (usually 30 min). The intensity of dyed bands in gel represents enzymatic activity. The proteins amount was detected by SDS–PAGE followed by Coomassie blue staining.

To test the effect of phosphorylation by TMK4 on TAA1 enzymatic activity, TAA1$^{WT/T101A}$ CDS were amplified and inserted into pGEX vector containing a N-terminal GST tag followed by a protease cleavage site. GST-TAA1 proteins were isolated from E. coli and further purified by Mono S cation exchange chromatography column (#GE17516801) after cleavage by tobacco etch virus protease. TAA1 proteins without any tag were mixed with E. coli-purified His-sumo-TMK4 kinase domains (578aa–928aa) in a kinase reaction buffer (50 mM HEPES, 10 mM MgCl$_2$, 10 mM MnCl$_2$, and 1 mM DTT) with or without ATP at 25 °C to perform a kinase reaction then followed by in-gel Trp-AT activity assay. The intensity of dyed bands in native gel was measured by Quantity One software (v462) and standardized by TAA1 protein amount.

**IPA measurement via Salkowski reagent.** To test TAA1 enzymatic activity in vivo, Salkowski reagent was used to quantify IPA production. This assay was performed as previous reported[26,27] with slight modification. Briefly, ~50 mg root tissues were ground up in liquid nitrogen, added into the same volume of extraction buffer (100 mM HEPES, pH 7.5, 250 mM sorbitol, 5 mM ß-mercaptoethanol, 0.5% (v/v) Triton X-100, and 0.1% (w/v) PMSF), mixed well and kept on ice for 20 min. The extracted samples were centrifuged at 15,800×g for 30 mins at 4 °C. Fifty microleter total proteins were mixed with reaction buffer (100 mM phosphate buffer, pH 8.0, 10 mM L-tryptophan, 3 mM 2-oxoglutarate, 10 mM PLP, prewarmed at 37 °C for 3 mins). Proteins mixed with reaction buffer without adding α-ketoglutaric acid were used as a background control. The reactions were performed at 37 °C for 30 min before adding 1 mL Salkowski reagent (10 mM FeCl$_3$ and 35% [v/v] H$_2$SO$_4$). After incubation in the dark for 10 min at room temperature, the absorbance at 530 nm were measured by microplate reader (Thermo Variaskan Flash). Concentration of total protein was determined using Coomassie brilliant blue G250 dye by detecting the absorbance at 595 nm. The relative enzymatic activity was shown as ΔValue$_{530nm}$/mg total protein. This assay was also used to detect the enzymatic activity of E. coli-purified proteins. TAA1 CDS was amplified an inserted into PET14 vector and expressed in E. coli. Proteins were further purified by Ni-NTA resin (QIAGEN cat no. 30210). For the recombinant proteins, 2.5 μg TAA1-His proteins were used for each reaction then the absorbance was measured by Biotek cell imaging multimode reader (BioTek Instruments, Inc. Winooski, VT) at different time points.

**PLP detection by measuring absorbance at 388 nm.** PLP was reported to show maximum absorbance ~388 nm[46,47]. TAA1$^{WT}$-His, TAA1$^{T101A}$-His, TAA1$^{T101D}$-His, and TAA1$^{K217A}$-His proteins were purified form E. coli. A total of 600 μL proteins (2 μg/μL) were denatured at 95 °C for 5 mins to release PLP from proteins. The supernatant containing PLP was obtained by centrifuge and used for reading absorbance at 388 nm. Different concentrations of PLP (sigma, P9255) were set as positive controls.

**Yeast two-hybrid assay.** The C-terminals of TMK family proteins named TMK1C, TMK2C, TMK3C, and TMK4C[38] were cloned and inserted into the pGBKT7 vector as bait. The full-length CDS of D6PK, PINOID, and the kinase domains of BRI1, BAK1, and FLS2 were cloned and inserted into the pGBKT7 vector. TAA1 CDS was cloned, inserted into PDONR-Zeo then inserted into the pGWAD vector by LR reaction as prey. The Golden Yeast strain was used for yeast two-hybrid assay. Yeast transformation was done by the transformation kit (Frozen-EZ Yeast Transformation II Kit™, T2001, ZYMO RESEARCH), and the interaction was tested on -Trp-Leu and on -Trp-Leu-Ade-His medium.

**Coimmunoprecipitation assay in protoplast.** The coding region of TMK4 and TAA1 were amplified, and then inserted into the transient expression vectors 35S-HBT-HA-NOS and 35S-HBT-Flag-NOS, respectively. TMK4-HA and TAA1-Flag were coexpressed in Col-0 protoplasts, while single expression was used as a negative control. Total proteins were extracted from the protoplasts with the extraction buffer (50 mM Tris-HCl (pH 7.5), 150 mM NaCl, 5 mM EDTA, 5% glycerol, 1% TritonX-100 with protease inhibitor, phosphatase inhibitor, and PMSF), then incubated with anti-Flag beads (Sigma cat. #A2220) at 4 °C for 4 h followed by two times of washing (50 mM Tris-HCl (pH 7.5), 150 mM NaCl, 5 mM EDTA, 5% glycerol, and 0.01% TritonX-100). Proteins were separated on SDS–PAGE gels and detected by anti-HH (HA-7, H6533, Sigma, 1:2000 dilution) or anti-Flag (M20008L, Abmart, 1:2000 dilution) antibodies via western blots. 35S-TAA1-Flag and 35S-TAR1-GFP were generated to detect interaction between TAA1 and TAR1 using similar assay. TAR1-GFP were detected by an anti-GFP antibody (HT801, TransGen Biotech, 1:2000 dilution) via western blot.

**In vitro pull-down assay.** TAA1-His and His-Sumo proteins were expressed in E. coli and captured by Ni-NTA resin. TMK4 KD (587aa–928aa) sequence was amplified and inserted into pGEX4T-2 vector. GST-TMK4 KD proteins were purified from E. coli by Glutathione agarose (sigma G4510) and incubated with

TAA1-His or His-Sumo beads in binding buffer (20 mM HEPES, pH 7.5, 40 mM KCl, 1 mM EDTA, 1% glycerol, 0.1‰ TritonX-100 with PMSF) for 2 h at 4 °C. The beads were washed three times by washing buffer (20 mM HEPES, pH 7.5, 40 mM KCl, 1 mM EDTA, and 0.05‰ TritonX-100). GST-TMK4 KD proteins were detected by GST antibody (M20007, Abmart, 1:2000 dilution) via western blots, and His-TAA1 and His-Sumo proteins were detected by Coomassie blue staining.

**In vitro kinase assay.** TMK4 kinase domain(578aa–928aa) was inserted into His-sumo vector. His-sumo-TMK4KD proteins and TAA1-His proteins were mixed in the reaction buffer (50 mM HEPES, 10 mM MgCl$_2$, 10 mM MnCl$_2$, 1 mM DTT with 50 μM γ$^{32}$P ATP, and 50 μM ATP). After incubation at 25 °C for 1 h, the reaction was stopped by adding 5x SDS loading buffer. Radioactive signals from TAA1-His were detected using the Typhoon imaging system (GE Healthcare TYPHOON FLA 9500).

To detect phosphorylation sites of TAA1 targeted by TMK4 in vitro, similar assay was performed using TAA1-His or GST-TAA1 as substrate, His-sumo-TMK4KD as kinase with 50 μM cold ATP adding in reaction system. After phosphorylation reaction, the substrates were sent for mass spectrometric analysis to identify phosphorylation sites of TAA1 in vitro. TAA1 proteins without kinases adding were used as a negative control group.

**GUS staining and quantification.** Five-day-old seedlings were fixed in 90% acetone in ice for 20 min, washed by the staining buffer (100 mM NaH$_2$PO$_4$ pH 7.0, 10 mM EDTA, 0.5 mM K$_4$[Fe(CN)$_6$]·3H$_2$O, 0.5 mM K$_3$[Fe(CN)$_6$], 0.1% TritonX-100), and incubated in the staining buffer with 0.5 mg/mL 5-bromo-4-chloro-3-indolyl-b-d-glucuronic acid (x-Gluc) at 37 °C for 3 h. The reactions were stopped by adding 75% ethanol. Images were taken by Leica DM6B Microscope with roots immersed in clearing buffer (chloral hydrate: water: glycerol—8:3:1). For the quantification of GUS activity, total proteins were extracted from seedlings and used for GUS catalytic reaction using 4-methylumbelliferyl-β-D-glucuronide hydrate as substrate[48]. The products of GUS enzyme were detected by reading fluorescence absorbance (excitation: 365 nm, emission: 455 ± 5 nm) via microplate reader (Thermo Variaskan Flash). Relative GUS activity was standardized by total protein amount measured by Coomassie blue G250 dye.

**DII-Venus signal observation and quantification.** *35S-DII-Venus* marker line was crossed with *tmk4-2* to obtain *35S-DII-Venus;tmk4-2*. Five-day-old seedlings were used for Venus signal observation detected by Leica SP8 confocal microscope (Leica TCS SP8 X, excitation: 514 nm, emission: 520–550 nm) under a Z-stack condition. A fixed oblong region with a fixed distance to QC (quiescent center) cells in the root meristem was selected and the overall signals in this fixed region, including all nuclear signal were quantified by ImageJ software (v1.8.0). Fluorescence intensity per unit area was used for statistical comparison.

**Phenotypic observation of *Marchantia*.** Transgenic plants of *Marchantia* were obtained through agrobacterium-mediated thallus transformation[49]. Thalli grown on 1/2 B5 medium containing dimethyl sulfoxide (DMSO) or 500 μM L-Kyn (sigma K3750) for 16 days were used for phenotypic observation. The area of thallus was measured by image J software (v1.8.0) to represent the thallus size.

**Gel shift assay.** Five-day-old seedlings were collected and ground up in liquid nitrogen. Proteins were extracted by 2x SDS loading buffer (100 mM Tris-HCl pH 6.8, 4% SDS, 12% glycerol, 0.02% bromophenol blue, and 2% ß-mercaptoethanol) followed by 95 °C denaturation. Proteins were separated on 6% SDS–PAGE gel containing 50 μM phos-tag (Phos-tag $^{TM}$ Acrylamide AAL-107, WAKO) and 100 μM MnCl$_2$ and detected by anti-TMK4 antibody (135-360aa, generated by ABclonal Biotechnology, 1:2000 dilution).

**Structure analysis.** Information of the TAA1 protein structure (PDB code: 3bwn) was obtained from the PDB database (http://www.rcsb.org/)[9]. Pymol (Version 0.99rc6) software was used for structure analysis and image capture.

**Phylogenetic analysis of TAA1 family.** Information of protein sequences was obtained from the NCBI database (https://www.ncbi.nlm.nih.gov/) and the phytozome database (https://phytozome.jgi.doe.gov/pz/portal.html). Clustal Omega was used for sequence alignment (https://www.ebi.ac.uk/Tools/msa/clustalo/).

**Statistical analysis.** All box plots and bar graphs were generated by GraphPad Prism 7 software. To compare multiple groups, analysis of variance (ANOVA) one-way test was applied using the GraphPad Prism 7 software. All $P$ values which is <0.0001 were shown as $P < 0.0001$, $P > 0.05$ indicates no significant difference. Lines in dot plots indicate means with s.d.

**Reporting summary.** Further information on research design is available in the Nature Research Reporting Summary linked to this article.

## Data availability

The data supporting the findings in this study are available and described within the paper and its supplemental information. Source Data (gels and graphs) for Figs. 1–4 and Supplementary Figs. 2–6, 8, 10–14 are provided as a separate Source Data file.

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

## Acknowledgements

We thank J. Du for providing His-sumo vector. We also thank T. Kohchi and C. Yamamuro for providing *M. polymorpha* (Tak-1) materials and transformation protocol. This work was supported by the National Natural Science Foundation of China (grant 31422008 and 31870256), the National Key R&D Program of China (2016YFA0503200), and the start-up funds from PSC and FAFU to T.X.

## Author contributions

T.X. and Q.W. initiated this project and designed the experiments. Q.W. and G.Q. carried out most of the experiments. M.C. helped with the in vitro kinase assay, R.C. helped with phenotypic analysis, Y.H. assisted with TMK4 phosphorylation assay, L.Y. and Z.Z. generated MpTAA constructs, Y.Y. helped to make TMK4 antibodies, and Y.G. assisted with protein purification. W.X. helped with the computer analysis of the TAA1 structure, and A.T. helped with mass spectrometric analysis. T.X. and Q.W. wrote the manuscript.

## Competing interests

The authors declare no competing interests.
