## [Peer Review File · Nature Communications]

Reviewers' comments:

Reviewer #1 (Remarks to the Author):

Auxin is a key plant hormone that regulates multiple aspects of plant growth and development. The generation of local auxin gradients consisting of a balance between auxin transport and auxin biosynthesis is a crucial aspect of the determination of developmental fate of plant organs. Here the authors have explored the molecular basis of on-off regulation of a key auxin biosynthetic enzyme TAA1. Using mass spectrometry they have demonstrated that phosphorylation of a Threonine residue at position 101 regulates the enzymatic activity of the TAA1 protein. Using the Arabidopsis root as a tool, they have shown that phosphorylation of T101 in TAA1 results in a switched off state leading to lower auxin levels in the root meristem and shorter root hairs. Furthermore, they demonstrate that the T101 site is evolutionarily conserved across the plant kingdom and a similar TAA1 phosphorylation-dependent mechanism acts in *Marchantia*. Lastly, using immunoprecipitation and pull down assays, they have identified that the TMK4 kinase that interacts with and phosphorylates TAA1 in vivo.

In general, the manuscript is concise and clear with experiments being well designed and of a high standard. As far as I am aware, this is the first time that a post translation modification has been shown to play such an important role in regulating auxin biosynthesis, making this work novel. The data is statistically analysed well and in particular the box plots showing individual data points for each experiment are to be commended. The materials and methods are written clearly and in detail and should enable other researchers to perform these and similar experiments with a high degree of reproducibility. However the following concerns need to be addressed prior to acceptance in *Nature Communications*.

Major

1. Lines 104 – 115: The authors have generated phosphomimic / nonphosphorylatable mutations in TAA1 and described how the TAA1 T101D mutation abolishes enzymatic activity. Even though they are repeating previously described methods, there needs to be more description of the method and the expected outcome in this section to describe the lack of bands in Figure 1c. I am aware that they have provided this in the methods section, however it would make it much clearer to include a brief description in the text and legend for Figure 1c.
2. A concern I have for figure 1c is that the lane for T101D looks like it has been edited digitally. This is because of the sharp contrast between the lanes of T101D and other samples, as well as the noticeable absence of background in the T101D lane. This gel may simply have been edited for contrast or sharpness, however, the original gels should be provided and this image could be replaced with a less edited one.
3. Throughout the text, the authors have described a number of experiments performed with the *wei8-3 tar2-1* double mutant including transformation, measurement of meristem size and root hair length. From previously published work (Stepanova et al), The *wei8 tar2* double mutant is heterozygous for one of these genes (the double mutant is lethal) and in this double mutant the meristem was shown to terminate and eventually collapse in 5-6 days. The authors must explain these results and genotype the *wei8 tar2* mutant they have if they do not see these phenotypes.
4. Lines 164-168 and extended figure 7: The authors claim that a portion of TAA1-RFP is targeted to the plasma membrane where it interacts with TMK4 and does not seem to have any cytosolic localisation. However, in extended figure 7, TAA1 looks to be targeted to the plasma membrane only in epidermal cells and is cytosolic in cortical and other cells. While not impossible, the targeting of a single protein to completely different subcellular locations in different cell types is unusual at the very least. Given that the authors claim that TMK4 and TAA1 interact is a crucial part of this work, it would be nice to see additional images of TAA1 RFP and TMK4 GFP in the complete meristem.
5. In figure 2K, there seems to be no difference to thallus size between WT *Marchantia* and transgenic *Marchantia* containing TAA T224D both with DMSO and L-Kyn treatment. If there is, the authors should state this specifically and provide p values.

6. In Fig 4C, I have the same concern with the upper two extreme left gel lanes – the absence of any background. Again, these images should be replaced and the original gel images provided.
7. In the imaging of DII VENUS signal, the authors should describe which cell and/or tissue type was used for quantification, as DIIV is differentially expressed in trichoblast and atrichoblast cells in the epidermis. They should also describe if nuclear fluorescence was quantified and if the 'n' includes number of cells or number of roots. If the latter, then how many cells were measured in each root?
8. The authors have provided a model stating that phosphorylation of T101 provides a developmental and environmental switch to regulation of local auxin levels in the root meristem. However, I do not see any evidence to support the environmental effect on T101 phosphorylation. It would be nice to see some evidence of differential phosphorylation of T101 in TAA1 in some environmental conditions known to result in high auxin levels, for example high temperature. Alternately it could be nice to also see if increased exogenous IAA/NAA treatment results in increased T101 phosphorylation as a mechanism to reduce endogenous auxin levels.

Minor corrections:

1. Line 79 – 'Adapt the environmental change' to 'Adapt to environmental change'
2. Line 211 – 'TMK phosphorylation level increased' might be more clearly described as 'TMK4-mediated phosphorylation of TAA1 increased'.
3. Line 471 – 'Site mutation' to 'site directed mutagenesis'. Also include kit details and primer sequences
4. Line 474 – Entry vector pDONR. Which pDONR?
5. Line 496 – 'plant' to 'plants'
6. Line 542 – 'shake' to 'shaken'
7. Line 605 – Heading should be 'Co-Immunoprecipitation assay in protoplasts'
8. Line 619 – Correct spelling of 'inserted'
9. Line 638 – Change 'send' to 'sent'
10. Line 764 – 'Turkey' to 'Tukey'

Reviewer #2 (Remarks to the Author):

This is an interesting manuscript where the authors propose that the activity of the auxin biosynthetic enzyme TAA1 is regulated by phosphorylation via TMK4. Although the authors present experimental evidence that TAA1 is phosphorylated in vivo at a T101 and that T101D phosphomimic mutation abolishes the activity of this enzyme, it is less clear to me that the physiological significance of the phosphorylation of this amino acid in vivo has been convincingly proven. For example, the authors indicate that the TAA1-T101D form cannot rescue the phenotypic defects of the *wei8-3* mutant, whereas the T101A version and the WT rescue this mutant equally well (figure 1e and f). In my opinion, this argues against a significant regulatory role for this phosphorylation in the activity of TAA1. In the *wei8-3* TAA1-T101A, the regulation of TAA1 by phosphorylation should be abolished and therefore it should display auxin-related phenotypes. This is even more relevant when the authors argue that the enlarged meristem of *tmk4* mutant is due to the lack of phosphorylation of TAA1. If that was the case, the meristem of *wei8-3* TAA1-T101A should be similar to that of the *tmk4* mutant and much larger than that of the *wei8-3* TAA1-WT or WT plants, but that is not the case (figure 1f). I believe this is an important point that needs to be further explored. The authors should examine in more detail the phenotype of the single *wei8-3* as well as double *wei8-3 tar2-1* complemented with the TAA1p:TAA1-T101A and compare those phenotypes with those of the corresponding lines complemented with the WT genomic TAA1.

The authors present additional lines of evidence in support of the claim that the activity of TAA1 is regulated by phosphorylation via TMK4. Although the results presented in this regard are consistent with their hypothesis, I do not believe the data are sufficiently conclusive and could be

interpreted differently than the way the authors did. For example, the authors show that the high auxin levels observed in *mapk4*, bigger meristem and high auxin reporter activity can be suppressed by the *wei8-3* mutation. Although these results indicate that mutation in *tmk4* results in high levels of auxin production, the fact that *wei8-3* suppresses this phenotype does not provide support for TMK4 regulating TAA1 activity. I believe similar results would be obtained if the right combination of YUCCA mutants would have been used instead of *wei8-3*. In other words, *wei8-3* should suppress any auxin overproducing mutant that does so through the IPyA pathway independently of the mechanism by which the auxin production is activated.

Another, in my opinion, circumstantial evidence in favor of the proposed role of the T101 phosphorylation is the sequence conservation of this amino acid, but as it can be seen in figure 2 and extended figure 5, this strong conservation is also true for many other amino acids in this part of the protein (the PLP binding pocket). In light of the sequence conservation around T101, it is not surprising that the T101D substitution causes a loss of TAA1 activity, irrespective of the specific function T101 fulfills or the modification it carries.

A critical experiment linking the phosphorylation of TAA1 and its enzymatic activity is shown in figure 4e where the incubation of TAA1 with the TMK4 kinase results in the inactivation of TAA1. Since the authors have shown that TMK4 can interact with TAA1, this finding does not necessarily mean that the inactivation is due to phosphorylation. The authors should show that there is a correlation between the phosphorylation levels of TAA1 and its activity even after the TMK4 has been removed from the mixture.

If the authors' hypothesis is correct and TAA1 phosphorylation plays an important role in the regulation of this enzyme's activity, one would expect that the phenotype of the *wei8-3* mutant complemented with the TAA1-T101A transgene should be stronger (larger meristem) than that of the *tmk4* mutant where, according to the results in figure 4d, TAA1 is still phosphorylated in some degree (perhaps by some of the other TMK family members). The possible explanation of these results that would still agree with the authors' hypothesis is that the phenotype of *tmk4-1* mutant could be due to the activation in this mutant of TAR2 and TAR1. This possibility, however, does not seem plausible, since the enhanced IPyA production in the *mpk4-1* mutant is fully suppressed by the *wei8-3* mutation (figure 4f).

In summary, I find the possibility that TAA1 activity is regulated by phosphorylation via TMK4 very exciting. Unfortunately, most of the lines of evidence presented to support such hypothesis are not very conclusive and different interpretations are plausible. Specifically, I do not think the authors have provided sufficient evidence to claim that: 1) phosphorylation of TAA1 at T101 affects the enzymatic activity of this protein, and more importantly, 2) that the phosphorylation of TAA1 plays a physiologically relevant regulatory role. In fact, some of the results presented (the complementation of *wei8-3* by the TAA1-T101A mutant version of the gene) strongly argue against this possibility.

Point-by-Point Responses to Referees

Reviewer #1 (Remarks to the Author):

"Auxin is a key plant hormone that regulates multiple aspects of plant growth and development. The generation of local auxin gradients consisting of a balance between auxin transport and auxin biosynthesis is a crucial aspect of the determination of developmental fate of plant organs. Here the authors have explored the molecular basis of on-off regulation of a key auxin biosynthetic enzyme TAA1. Using mass spectrometry they have demonstrated that phosphorylation of a Threonine residue at position 101 regulates the enzymatic activity of the TAA1 protein. Using the Arabidopsis root as a tool, they have shown that phosphorylation of T101 in TAA1 results in a switched off state leading to lower auxin levels in the root meristem and shorter root hairs. Furthermore, they demonstrate that the T101 site is evolutionarily conserved across the plant kingdom and a similar TAA1 phosphorylation-dependent mechanism acts in Marchantia. Lastly, using immunoprecipitation and pull down assays, they have identified that the TMK4 kinase that interacts with and phosphorylates TAA1 in vivo.

In general, the manuscript is concise and clear with experiments being well designed and of a high standard. As far as I am aware, this is the first time that a post translation modification has been shown to play such an important role in regulating auxin biosynthesis, making this work novel. The data is statistically analysed well and in particular the box plots showing individual data points for each experiment are to be commended. The materials and methods are written clearly and in detail and should enable other researchers to perform these and similar experiments with a high degree of reproducibility. However the following concerns need to be addressed prior to acceptance in Nature Communications. "

Response: We greatly appreciate the encouraging comments and constructive suggestions from the referee. In the revised manuscript, we addressed all these concerns.

Major

"1. Lines 104 – 115: The authors have generated phosphomimic / nonphosphorylatable mutations in TAA1 and described how the TAA1 T101D mutation abolishes enzymatic activity. Even though they are repeating previously described methods, there needs to be more description of the method and the expected outcome in this section to describe the lack of bands in Figure 1c. I am aware that they have provided this in the methods section, however it would make it much clearer to include a brief description in the text and legend for Figure 1c."

Response: Thanks for pointing out this. As suggested, we added more descriptions of the enzymatic activity assay in the text (line 99-105) and figure legend for Figure 1c. In brief, the *E.coli*-purified TAA1 proteins were separated by a native gel and followed by the transaminase catalytic reaction and further staining. The intensity of bands in gel represented for enzymatic activity. Lack of dark band in GST-TAA1^{T101D} lane indicated that TAA1^{T101D} totally lost enzymatic activity.

"2. A concern I have for figure 1c is that the lane for T101D looks like it has been edited digitally. This is because of the sharp contrast between the lanes of T101D and other samples, as well as the noticeable absence of background in the T101D lane. This gel may simply have been edited for contrast or sharpness, however, the original gels should be provided and this image could be replaced with a less edited one."

Response: Thanks the referee for this point. As suggested, we replaced the previous image with the less edited one and provided the original images. We performed a chemical reaction for transaminase enzyme directly in the gel, and the lane with TAA1^{T101D} proteins showed white colour after catalytic reaction since TAA1^{T101D} lost enzymatic activities. Thus, it would somehow look like sharp background in T101D lane (please refer to source data).

"3. Throughout the text, the authors have described a number of experiments performed with the wei8-3 tar2-1 double mutant including transformation, measurement of meristem size and root hair length. From previously published work (Stepanova et al), The wei8 tar2 double mutant is heterozygous for one of these genes (the double mutant is lethal) and in this double mutant the meristem was shown to

terminate and eventually collapse in 5-6 days. The authors must explain these results and genotype the *wei8 tar2* mutant they have if they do not see these phenotypes. ”

Response: As suggested, we genotyped the *wei8-3;tar2-1* mutant used in our manuscript as shown below. The *wei8-3;tar2-1* double homozygous mutant we used contains a *wei8-3* allele, different from the *wei8-1* allele used in the original paper (Stepanova et al, 2008). According to their paper, both *wei8-3* and *wei8-1* are strong mutants, but there is some expression of the truncated mRNA in the *wei8-3* allele as shown in the supplementary materials of Stepanova’s paper. It was reported the *wei8-1;tar1-1;tar2-1* triple mutants showed severe seedling lethal phenotype, while the *wei8-1;tar2-1* double homozygous mutants displayed weaker phenotype that the seedling can survive until florescence stage (Stepanova et al, 2008). The *wei8-3;tar2-1* mutant showed the similar but weaker root meristem defects than the reported *wei8-1;tar2-1*. Based on our examination, among 5-day-old *wei8-3;tar2-1* seedlings, around 64% showed obviously reduced root meristem, 32% showed strongly attenuated root meristem, and 3% displayed collapsed root meristem (Response Fig.1). As suggested by the referee, we added more descriptions about the mutant we used in the revised manuscript (line118, supplementary Fig.3 and methods).

Response Fig. 1

Response Figure 1. Characterization of *wei8-3;tar2-1* mutant

(a) Description of *wei8;tar2* transfer DNA (T-DNA) insertion mutants. Lines represent introns; black and grey boxes represent exons and untranslated regions. *wei8-3* (Salk_127890), *wei8-1* (CS_31113-19), *tar2-1* (Salk_021258), *tar2-2* (Salk_137800). Arrows show primers used for genotyping.

(b) Genotyping results of *wei8-3;tar2-1* double mutant. Col-0 was set as control.

(c) Attenuate root meristem phenotype of *wei8-3;tar2-1*. Pictures were shown with 5-day-old roots. The ratio of different root meristem phenotypes was shown as indicated. n represents number of seedlings used for quantification. Scale bar, 75 μ m.

“4. Lines 164-168 and extended figure 7: The authors claim that a portion of TAA1-RFP is targeted to the plasma membrane where it interacts with TMK4 and does not seem to have any cytosolic localisation. However, in extended figure7, TAA1 looks to be targeted to the plasma membrane only in epidermal cells and is cytosolic in cortical and other cells. While not impossible, the targeting of a single protein to completely different subcellular locations in different cell types is unusual at the very least. Given that the authors claim that TMK4 and TAA1 interact is a crucial part of this work, it would be nice to see additional images of TAA1 RFP and TMK4 GFP in the complete meristem.”

Response: Thanks the referee for this excellent point. The subcellular co-localization determines the possible biochemical link between two proteins. We might cause the misunderstanding in our previous version of manuscript that TAA1 was targeted to plasma membrane. TAA1 is a cytosolic localized protein. In cortical cells and distal stem cells, where TAA1 expression is high, we can see clearly cytosolic distribution. In other type of cells like epidermal cells, TAA1 expression is low and may be squeezed by the vacuoles to the sub-membrane regions which looks like the plasma membrane localization but actually it is still cytosolic distribution. As suggested, we added additional images to show the co-localization between TAA1 and TMK4 in the complete root meristem in the revised manuscript (supplementary Fig. 9). These images are consistent with those shown in the original manuscript. From these images, we proposed that a portion of cytosolic TAA1 proteins were targeted by TMK4 at the sub-membrane region, which was also supported by the Co-IP method. In the revised manuscript, we changed the description to clarify the localization of TAA1 (line 182-185).

"5. In figure 2K, there seems to be no difference to thallus size between WT Marchantia and transgenic Marchantia containing TAA T224D both with DMSO and L-Kyn treatment. If there is, the authors should state this specifically and provide p values."

Response: Thanks the referee for this comment. We agreed with the referee that we should provide the statistical analysis with *P* values for these data. By using two-sided *t*-test, we found the thallus size in wild type was weakly but significantly different from the 35S-MpTAA^{T224D} transgenic lines when treated with L-Kyn chemical (*P*-value: 0.0389) (Fig. 2k). We added this information in both the figure and text (line 167).

"6. In Fig 4C, I have the same concern with the upper two extreme left gel lanes – the absence of any background. Again, these images should be replaced and the original gel images provided."

Response: Thanks for pointing out this. As suggested, we provided the original gel data and replaced the figure here (revised Fig. 3d). Actually, the absence of the background was caused by the extremely strong signal in the neighbouring lanes (please refer to source data).

"7. In the imaging of DII VENUS signal, the authors should describe which cell and/or tissue type was used for quantification, as DIIV is differentially expressed in trichoblast and atrichoblast cells in the epidermis. They should also describe if nuclear fluorescence was quantified and if the 'n' includes number of cells or number of roots. If the latter, then how many cells were measured in each root?"

Response: Thanks the referee for these valuable suggestions. We have added more detailed descriptions of the quantification of DII-Venus signal in the methods section (line 615-618). In brief, we selected a fixed region in root meristem with the fixed distance to the QC cells, and quantified the overall signal in this region including nuclear signal. "n" denotes the number of roots and this information has been added in the figure legend (supplementary Fig. 13b). The root images were taken under a Z-stack condition so it was hard to calculate the accurate cell number. Fluorescence intensity per unit area was calculated by Image J software, and used for statistical analysis.

"8. The authors have provided a model stating that phosphorylation of T101 provides a developmental and environmental switch to regulation of local auxin levels in the root meristem. However, I do not see any evidence to support the environmental effect on T101 phosphorylation. It would be nice to see some evidence of differential phosphorylation of T101 in TAA1 in some environmental conditions known to result in high auxin levels, for example high temperature. Alternately it could be nice to also see if increased exogenous IAA/NAA treatment results in increased T101 phosphorylation as a mechanism to reduce endogenous auxin levels."

[redacted]

To test whether the exogenous auxin treatment would change T101 phosphorylation of TAA1 proteins, we treated the protoplast expressed TAA1-GFP with 500 nM IAA for 10 mins. After

precipitated TAA1-GFP from protoplast, we performed quantitative mass spectrometric analysis and observed an increased phosphorylation level of T101 on TAA1 proteins (supplementary Fig. 14b), indicating that there might be a feedback regulation of auxin levels by this non-transcriptional regulation of auxin biosynthesis, which is important for auxin homeostasis. This mechanism is required for root development including root meristem and root hair based on the clear phenotypes in the *TAA1^{T101D};wei8-3* and *tmk4* mutants.

Therefore, we conclude that this non-transcriptional regulation of auxin biosynthesis plays an important role in the environmental and developmental regulation of auxin levels.

Minor corrections:

1. *“Line 79 – ‘Adapt the environmental change’ to ‘Adapt to environmental change’”*

Response: Thanks for pointing out this mistake. We have changed accordingly in the text (revised text line 69).

2. *“Line 211 – ‘TMK phosphorylation level increased’ might be more clearly described as ‘TMK4-mediated phosphorylation of TAA1 increased.’”*

Response: We noticed that the referee might misunderstand this data. Here we want to show that TMK4 phosphorylation level was induced by auxin (revised text line 232).

3. *“Line 471 – ‘Site mutation’ to ‘site directed mutagenesis’. Also include kit details and primer sequences”*

Response: Thanks for the advice. We have changed the text accordingly, and included the kit details and primer information in the methods section (revised text line 420-422).

4. *“Line 474 – Entry vector pDONR. Which pDONR?”*

Response: It is pDONR-zeo. We have added the information in the text (revised text line 424).

5. *“Line 496 – ‘plant’ to ‘plants’”*

Response: We have changed according to the suggestion (revised text line 445).

6. *“Line 542 – ‘shake’ to ‘shaken’”*

Response: We have changed according to the suggestion (revised text line 496).

7. *“Line 605 – Heading should be ‘Co-Immunoprecipitation assay in protoplasts’”*

Response: We have changed the heading as suggested (revised text line 566).

8. *“Line 619 – Correct spelling of ‘inserted’”*

Response: We have corrected the spelling mistake here (revised text line 581).

9. *“Line 638 – Change ‘send’ to ‘sent’”*

Response: We have corrected the spelling mistake here (revised text line 599).

10. *“Line 764 – ‘Turkey’ to ‘Tukey’”*

Response: We have corrected the spelling mistake here (revised text line 877).

Again, we appreciate all suggestions from this referee that help us to improve our manuscript, which is more readable to the audience.

Reviewer #2 (Remarks to the Author):

"This is an interesting manuscript where the authors propose that the activity of the auxin biosynthetic enzyme TAA1 is regulated by phosphorylation via TMK4. Although the authors present experimental evidence that TAA1 is phosphorylated *in vivo* at a T101 and that T101D phosphomimic mutation abolishes the activity of this enzyme, it is less clear to me that the physiological significance of the phosphorylation of this amino acid *in vivo* has been convincingly proven. For example, the authors indicate that the TAA1-T101D form cannot rescue the phenotypic defects of the *wei8-3* mutant, whereas the T101A version and the WT rescue this mutant equally well (figure 1e and f). In my opinion, this argues against a significant regulatory role for this phosphorylation in the activity of TAA1. In the *wei8-3* TAA1-T101A, the regulation of TAA1 by phosphorylation should be abolished and therefore it should display auxin-related phenotypes. This is even more relevant when the authors argue that the enlarged meristem of *tmk4* mutant is due to the lack of phosphorylation of TAA1. If that was the case, the meristem of *wei8-3* TAA1-T101A should be similar to that of the *tmk4* mutant and much larger than that of the *wei8-3* TAA1-WT or WT plants, but that is not the case (figure 1f). I believe this is an important point that needs to be further explored. The authors should examine in more detail the phenotype of the single *wei8-3* as well as double *wei8-3 tar2-1* complemented with the TAA1p:TAA1-T101A and compare those phenotypes with those of the corresponding lines complemented with the WT genomic TAA1."

Response: Thanks for the constructive and valuable comments from this referee. We followed the referee's suggestions to further improve our manuscript by adding new data and more descriptions. To illustrate the physiological role of this phosphorylation-dependent regulation of TAA1 enzyme, we tested a couple of hypotheses, which was described in the responses to referee #1. Briefly, we found that environmental changes such as heat shock regulated auxin levels partially through this non-transcriptional regulation of auxin biosynthesis enzyme (Response Fig. 2). In addition, we also found that direct auxin treatment increased the T101 phosphorylation level of TAA1 proteins indicating a self-regulation mechanism of auxin concentration in plants (supplementary Fig.14b). All these data indicated that this phosphorylation-based regulation mechanism of auxin biosynthesis might participate in both the environmental responses and the feedback regulation of auxin concentration.

We also agreed with the referee that TAA1^{T101A} phenotype needs to be further analysed. As suggested by the referee, in the revised manuscript, we designed a couple of new experiments to test the function of T101A mutation on TAA1 proteins:

1) As reported (Stepanova et al, 2008), *E.coli*-purified TAA1 proteins could carry certain amount of PLP co-factor from bacterial and catalyse tryptophan to IPA without additional PLP supply. We found that compared with *E.coli*-purified TAA1^{WT}, the TAA1^{T101A} proteins did not have any enzymatic activity without exogenous PLP as TAA1^{T101D} proteins (supplementary Fig. 2a). Both TAA1^{T101D} and TAA1^{T101A} showed weaker binding with PLP (supplementary Fig. 2c). However, if we supplied sufficient exogenous PLP, the TAA1^{T101A} proteins but not TAA1^{T101D} proteins started to show the enzymatic activity even higher than TAA1^{WT} proteins (Fig. 1d; supplementary Fig. 2b). This suggested that A was not able to fully simulate non-phosphor mimic of T, TAA1^{T101A} proteins were able to react with PLP co-enzyme but their ability to capture PLP was reduced.

2) Suggested by the referee, we further analysed phenotypes in different TAA1^{T101A}; *wei8-3* alleles and TAA1^{T101A}; *wei8-3*; *tar2-1*, and found TAA1^{T101A} only partially complemented *wei8-3* or *wei8-3*; *tar2-1* phenotype in root meristem size (supplementary Fig. 5a-d). But when supplied with 5 μ M PLP, TAA1^{T101A}; *wei8-3* increased root meristem cell number and even showed the phenotype as in the *tmk4* mutant (Response Fig. 3), consistent with the increased enzymatic activity of TAA1^{T101A} proteins when supplied with sufficient exogenous PLP.

Taken together, TAA1^{T101A} was not fully functional *in vivo* due to its abnormal ability to capture PLP co-factor. Although, the amino acid alanine was normally used to simulate the non-phosphorylated version of amino acid, but it did not always function well (Joanna, 2012; Yves, 2019; Arne, 2016) which might be due to that the structure of alanine was not absolutely equal to the structure of non-phosphorylated threonine. These might explain why we did not see obvious auxin-related phenotype of TAA1^{T101A} transgenic plant. We have added the description and new data in the revised manuscript.

Response Fig. 3

Response Figure 3. PLP supply increased *TAA1^{T101A};**wei8-3* root meristem size.

Quantification of root meristem size treated with 5 µM PLP. 5-day-old seedlings grown on 1/2 MS plates with or without 5 µM PLP were used for phenotype analysis. *P*-values were shown as indicated. *n* represents number of seedlings. Two-sided *t*-test. Three repeats with similar results.

"The authors present additional lines of evidence in support of the claim that the activity of TAA1 is regulated by phosphorylation via TMK4. Although the results presented in this regard are consistent with their hypothesis, I do not believe the data are sufficiently conclusive and could be interpreted differently than the way the authors did. For example, the authors show that the high auxin levels observed in *mapk4*, bigger meristem and high auxin reporter activity can be suppressed by the *wei8-3* mutation. Although these results indicate that mutation in *tmk4* results in high levels of auxin production, the fact that *wei8-3* suppresses this phenotype does not provide support for TMK4 regulating TAA1 activity. I believe similar results would be obtained if the right combination of YUCCA mutants would have been used instead of *wei8-3*. In other words, *wei8-3* should suppress any auxin overproducing mutant that does so through the IPyA pathway independently of the mechanism by which the auxin production is activated."

Response: We agreed with the referee that the genetic data alone cannot conclude that TMK4 regulates auxin concentration via TAA1. Based on genetic evidence that *tmk4-1* phenotype was mostly restored by TAA1 mutation, we can only demonstrate that TAA1 or its downstream YUCCA based auxin biosynthesis contributes the majority of the overproduced auxin in *tmk4-1* mutant. However, besides the genetic data, we also showed multiple lines of biochemical data: 1) TMK4 and TAA1 biochemically interacted with each other both *in vitro* and *in vivo* (Fig.3a-c; supplementary Fig.9); 2) TMK4 phosphorylated TAA1 at T101 site both *in vitro* and *in vivo* (Fig.3d-e; supplementary Fig.10); 3) TMK4 regulated TAA1 enzymatic activity both *in vitro* and *in vivo* (Fig.3f; supplementary Fig.11); 4) Phosphorylation of TAA1 at T101 site affected TAA1 enzymatic activity followed by auxin biosynthesis (Fig. 1b-f). These solid biochemical data strongly corroborate our genetic data which support our hypothesis that TMK4 regulates TAA1 through the direct phosphorylation. In the revised manuscript, we have modified the description of genetic-based conclusion as the referee pointed out (line227-230 and discussion).

“Another, in my opinion, circumstantial evidence in favor of the proposed role of the T101 phosphorylation is the sequence conservation of this amino acid, but as it can be seen in figure 2 and extended figure 5, this strong conservation is also true for many other amino acids in this part of the protein (the PLP binding pocket). In light of the sequence conservation around T101, it is not surprising that the T101D substitution causes a loss of TAA1 activity, irrespective of the specific function T101 fulfills or the modification it carries.”

Response: This is an excellent point. We appreciate the strong logic from the referee to interpret the data. Indeed, the PLP binding pocket is well conserved during evolution since it is very important for TAA1 function. We agreed with the referee that it is not surprising to see that T101 residue is conserved and other amino acids in this region may also be very important for TAA1 function. However, our manuscript focused on the phosphorylation-based regulation of TAA1 proteins because we discovered the conserved existence of the T101 phosphorylation *in vivo* both in *Arabidopsis* and *Marchantia* (T224) by mass spectrometric analysis (Fig.1a; supplementary Fig.8c). Then we provided multiple lines of evidences to support that the phosphorylation of T101 residue is important for TAA1 function. To further address the referee’s concern, we tested additional two substitutions of T101. If the function of T101 residue is only due to the conserved sequence itself, all other mutations should have the similar effect on TAA1 function. However, the two non-phosphor mimic mutations A (Alanine) and V (Valine) did not alter TAA1 enzymatic activity when supplied with sufficient PLP (Response Fig.4), but the two phosphor mimic mutations D (Aspartic acid) and E (Glutamic Acid) totally blocked the enzymatic activity of TAA1 (Response Fig.4). This strongly supported that the phosphorylation at T101 is important for the regulation of TAA1, but not only sequence itself. Again, we really appreciate the referee for this great suggestion. We feel it might be a bit out of scope of the current focus of this manuscript, thus just provide the data in the response letter. However, if the referees strongly feel that we should include this part, we have the requested data ready to do so.

Response Fig. 4

Response Figure 4. The function analysis of different mutations at T101 site on TAA1 proteins

Transaminase catalytic reactions were performed in the reaction reagents containing 2.5 μg *E.coli*-purified recombinant TAA1-His proteins with different mutations at T101 site (A, V, D and E). IPA products were detected with Salkowski reagent by reading the absorbance at 530 nm at different reaction time points (see methods). Values denoted the means \pm SD (n=three biological independent repeats). The co-factor PLP (10 μM) was supplied in reaction system (a) but not in (b).

“A critical experiment linking the phosphorylation of TAA1 and its enzymatic activity is shown in figure 4e where the incubation of TAA1 with the TMK4 kinase results in the inactivation of TAA1. Since the authors have shown that TMK4 can interact with TAA1, this finding does not necessarily mean that the

inactivation is due to phosphorylation. The authors should show that there is a correlation between the phosphorylation levels of TAA1 and its activity even after the TMK4 has been removed from the mixture.”

Response: Again, we really appreciate the referee’s thoroughness. We noticed that the referee might misunderstand this experimental design due to the insufficient description in the original manuscripts, thus we revised accordingly. In Figure 3f (initial version Figure 4e), to exclude the possibility that the interaction between TMK4 kinase domain and TAA1 influenced the enzymatic activity, we added both proteins and used ATP to trigger the phosphorylation reactions, and then followed by the transaminase catalytic reaction. TAA1 enzymatic activity was reduced dramatically when added ATP compared with the control group without ATP. This result suggested that it was the phosphorylation that leads to the inactivation of TAA1 enzyme but not the interaction with TMK4. In the meanwhile, we also showed that enzymatic activity of TAA1^{T101A} was not affected by the phosphorylation reaction, again suggesting the correlation between phosphorylation levels of TAA1 and its activity. We added more descriptions in the text and figure legends (line196-198).

“If the authors’ hypothesis is correct and TAA1 phosphorylation plays an important role in the regulation of this enzyme’s activity, one would expect that the phenotype of the wei8-3 mutant complemented with the TAA1-T101A transgene should be stronger (larger meristem) than that of the tmk4 mutant where, according to the results in figure 4d, TAA1 is still phosphorylated in some degree (perhaps by some of the other TMK family members). The possible explanation of these results that would still agree with the authors’ hypothesis is that the phenotype of tmk4-1 mutant could be due to the activation in this mutant of TAR2 and TAR1. This possibility, however, does not seem plausible, since the enhanced IPyA production in the mpk4-1 mutant is fully suppressed by the wei8-3 mutation (figure 4f). ”

Response: We totally agreed with the referee that we should clarify the functional connection between TMK4 and TAA1. Firstly, as explained above, T101A mutation on TAA1 proteins did not fully mimic the non-phosphorylated TAA1 likely due to the strict requirement of the protein structure, which partially explained that the phenotype of TAA1^{T101A};wei8-3 was not as strong as tmk4-1. Secondly, we noticed that the phenotypes in tmk4-1 such as root meristem size, root hair and free IAA levels were mostly but not fully restored by wei8-3 mutant, suggesting that tmk4 phenotype is not solely due to the lost function of TAA1 (Fig. 4c, 4e; supplementary Fig.13c). However, as the referee pointed, the transaminase activity in tmk4-1 was fully suppressed by wei8-3, suggesting the other homologs of TAA1 family might not contribute a lot to the auxin accumulation in the tmk4-1 mutant. This suggested that TMK4 might have other downstream targets involved in regulation of auxin concentration besides TAA1 family. In the meanwhile, we agreed with the referee that TAA1 was still phosphorylated in the tmk4-1 mutant (Fig.3e) indicating TAA1 might be targeted by other kinases. We added more discussion about this part in the revised manuscript (please see discussion).

“In summary, I find the possibility that TAA1 activity is regulated by phosphorylation via TMK4 very exciting. Unfortunately, most of the lines of evidence presented to support such hypothesis are not very conclusive and different interpretations are plausible. Specifically, I do not think the authors have provided sufficient evidence to claim that: 1) phosphorylation of TAA1 at T101 affects the enzymatic activity of this protein, and more importantly, 2) that the phosphorylation of TAA1 plays a physiologically relevant regulatory role. In fact, some of the results presented (the complementation of wei8-3 by the TAA1-T101A mutant version of the gene) strongly argue against this possibility.”

Response: We thank the referee for his/her positive comments and helpful critiques. As described above, we have added new data and more descriptions to address the two major concerns in the revised manuscript. This is recapped as below:

- 1) We generated multiple complementary lines of evidence to support that the *in vivo* phosphorylation of TAA1 at T101 regulates the enzymatic activity of this protein: 1. The structure analysis indicated the phosphorylation at T101 site would obstruct TAA1 binding to cofactor PLP

(Fig.1b); 2. Phosphor mimic substitution of T101 totally lost enzymatic activity *in vitro* (Fig.1c,d); 3. Phosphor mimic mutation T101D cannot complement *wei8-3* phenotype suggesting a non-functional protein *in vivo* (Fig.1d-e); 4. Phosphor mimic mutation T101D enhanced *wei8-3;tar2-1* phenotype, indicating a dominant effect of TAA1^{T101D} *in vivo* (Fig.2a-h); 5. TMK4 kinase domain reduced TAA1 enzymatic activity through the phosphorylation at T101 residue (Fig.3). 6. The *tmk4-1* mutant exhibited increased TAA1 enzymatic activity and auxin levels, and these changes were mostly rescued by the *wei8-3* loss of TAA1 mutation (Fig.4a-e). We agree that any of these results alone cannot conclusively demonstrate the regulation of TAA1 activity by TMK4-mediated TAA1 phosphorylation, but all together indicate the alternative hypothesis is extremely unlikely.

- 2) For the concern about the phosphorylation of TAA1 plays a physiologically relevant regulatory role, we have done: 1. Phosphor mimic mutation T101D show strong inhibition of TAA1 both *in vitro* and *in vivo* suggesting an important regulation role of TAA1 phosphorylation (Fig.1c-e); 2. These mechanism is essential for the control of root development including root meristem and root hair according to the phenotypes in TAA1^{T101D}*wei8-3* and *tmk4-1* (Fig.1d-e; Fig.4d-e; supplementary Fig.4; supplementary Fig.12). 3. T101 phosphorylation was identified *in vivo* both in *Arabidopsis* and *Marchantia* indicating the evolutionary significance of this modification (Fig.1a; supplementary Fig.8c); [redacted] 5. T101 phosphorylation level on TAA1 proteins was increased when treated with exogenous auxin implying a self-regulation loop of auxin biosynthesis to achieve auxin homeostasis in plant development (Supplementary Fig.14).

Combined all together, we proved that the *in vivo* phosphorylation at T101 residue on TAA1 proteins plays an important role in the regulation of TAA1 enzymatic activity and further auxin biosynthesis. And this non-transcriptional regulatory mechanism of auxin biosynthesis has the physiological importance in plant developmental and environmental responses.

References:

Stepanova, A. N. et al. TAA1-mediated auxin biosynthesis is essential for hormone crosstalk and plant development. *Cell* 133, 177-191, doi:10.1016/j.cell.2008.01.047 (2008).

Joanna Soroka, S. K. W., Nina Mausbacher, Thiemo Schreiber, Klaus Richter, Henrik Daub and Johannes Buchner. Conformational Switching of the Molecular Chaperone Hsp90 via Regulated Phosphorylation. *Molecular cell* 45, 517-528, doi:10.1016/j.molcel.2011.12.031 (2012).

Yves Dondelinger, Tom Delanghe, D. P., Meghan A. Wynosky-Dolfi, Daniel Sorobetea, Diego Rojas-Rivera, Piero Giansanti, Ria Roelandt, Julia Gropengiesser, Klaus Ruckdeschel, Savvas N. Savvides, Albert J.R. Heck, Peter Vandenabeele, Igor E. Brodsky & Mathieu J.M. Bertrand Serine 25 phosphorylation inhibits RIPK1 kinase-dependent cell death in models of infection and inflammation. *Nature communication* 10, doi:10.1038/s41467-019-09690-0 (2019).

Arne Ittner, S. W. C., Josefine Bertz, Alexander Volkerling, Julia van der Hoven, Amadeus Gladbach, Magdalena Przybyla, Mian Bi, Annika van Hummel, Claire H. Stevens, Stefania Ippati, Lisa S. Suh, Alexander Macmillan, Greg Sutherland, Jillian J. Kril, Ana P. G. Silva, Joel P. Mackay, Anne Poljak, Fabien Delerue, Yazhi D. Ke, Lars M. Ittner. Site-specific phosphorylation of tau inhibits amyloid- β toxicity in Alzheimer's mice. *Science* 354, 904-908 (2016).

REVIEWERS' COMMENTS:

Reviewer #1 (Remarks to the Author):

In general, the authors have done a great job to address my comments and concerns. The changes they have made have added clarity to the manuscript and broadly improved readability. The additional experimental detail and source data they have provided are of a high standard. I have a few minor comments and recommended changes below:

1. For comment number 4 regarding targeting of TAA1-RFP in supplementary figure 9, have the roots been counterstained with PI? If this is the case, this should be stated in the figure legend. Also a brief description of the construct (35S-TAA1-RFP? pTAA1-TAA1-RFP?) in the figure legend and the manuscript text would be helpful.

2. For comment number 8 regarding the environmental regulation of TAA1 phosphorylation: The authors have generated a nice dataset showing how temperature regulates TAA1 phosphorylation levels. It is up to the authors if they wish to include this in the manuscript as it stands. I understand that this data may be out of the scope of this current manuscript and may form the basis of a future study. However, if they do not wish to include this data currently, they should tone down their description of TAA1 regulation by environmental factors and should discuss this more speculatively.

3. The authors may wish to have their additional highlighted text checked for language and grammar by a native speaker.

Reviewer #2 (Remarks to the Author):

I think the authors have done an excellent job addressing all my questions and concerns. The finding that high temperature and exogenous auxin produce a significant change in the phosphorylation levels of TAA1 and that this correlates with a decrease in the overall levels of auxin (at least in the case of the temperature treatment) is an important addition to the manuscript. In my opinion, this information should be presented to the readers. My only comment in this regard is that I am not sure why the authors decided to use heat-shock (38C) instead of the classical high temperature (28-29C) treatment. The authors should cite previous work where such high temperatures were used to repress auxin biosynthesis. My other main concern was the lack of phenotypic differences between the *wei8* complemented with the WT or the T101A mutant. The authors now provide a good explanation for the original results by showing that the T101A mutation not only abolishes phosphorylation, but also leads to a reduction in the affinity for PLP. In fact, they observe that when the *wei8-3* TAA1 T101A plants are supplemented with PLP, their root meristem is significantly larger than that of the control and more similar to that of the *tmk4-1* mutant. I also find these results interesting enough to be included in this manuscript.

Overall, I believe this manuscript provides convincing lines of evidence for the regulation of TAA1 activity by phosphorylation, and that this phosphorylation is mediated by TMK4. In my opinion, these are important findings that have a significant impact on our current understanding of the regulation of auxin biosynthesis and will have important implications in shaping future research in this area.

Point-by-Point Responses to Referees

Reviewer #1 (Remarks to the Author):

"In general, the authors have done a great job to address my comments and concerns. The changes they have made have added clarity to the manuscript and broadly improved readability. The additional experimental detail and source data they have provided are of a high standard. I have a few minor comments and recommended changes below:"

Response: We greatly appreciate the encouraging comments from the referee.

"1. For comment number 4 regarding targeting of TAA1-RFP in supplementary figure 9, have the roots been counterstained with PI? If this is the case, this should be stated in the figure legend. Also a brief description of the construct (35S-TAA1-RFP? pTAA1-TAA1-RFP?) in the figure legend and the manuscript text would be helpful."

Response: Thanks for pointing out this. Actually, we did not stain the roots with PI, and only observed GFP and RFP signal. We gave the description of the constructs in the figure legends (Supplementary Fig. 9) that we used the transgenic plants driven by native promoters (*pTMK4-TMK4-GFP* crossed with *pTAA1-TAA1-RFP*). As requested, we also added the information in the revised manuscript text [redacted]

"2. For comment number 8 regarding the environmental regulation of TAA1 phosphorylation: The authors have generated a nice dataset showing how temperature regulates TAA1 phosphorylation levels. It is up to the authors if they wish to include this in the manuscript as it stands. I understand that this data may be out of the scope of this current manuscript and may form the basis of a future study. However, if they do not wish to include this data currently, they should tone down their description of TAA1 regulation by environmental factors and should discuss this more speculatively. "

Response: Thanks for supporting us. We agreed with the referee that the data about heat stress is very good basis for future study, and we preferred to publish this after we figure out the detail mechanism and related biological significance about this heat stress induced auxin level adjustment. This will be a new independent manuscript which is a bit out of the scope of this current manuscript. As suggested, we toned down the description of TAA1 regulation by environmental factors and mentioned this only in the Discussion.

"3. The authors may wish to have their additional highlighted text checked for language and grammar by a native speaker. "

Response: Thanks for your concern. We have requested a professional native speaker to check the language and grammar in our manuscript.

Reviewer #2 (Remarks to the Author):

"I think the authors have done an excellent job addressing all my questions and concerns.

The finding that high temperature and exogenous auxin produce a significant change in the phosphorylation levels of TAA1 and that this correlates with a decrease in the overall levels of auxin (at least in the case of the temperature treatment) is an important addition to the manuscript. In my opinion, this information should be presented to the readers. My only comment in this regard is that I am not sure why the authors decided to use heat-shock (38C) instead of the classical high temperature (28-29C) treatment. The authors should cite previous work where such high temperatures were used to repress auxin biosynthesis."

Response: Thanks a lot for pointing out this. Global warming is a hot topic and previous studies have showed heat stress (around 37°C or higher) affects plant development especially for crop yield (Gourdji, S.M. et al. 2013; Wu, C. et al. 2019). Heat stress has been reported to reduce auxin level in

plants (Sharma L. et al. 2018; Wu, C. et al. 2019). This is different from the warm temperature (28-29°C)-mediated thermal morphogenesis during which auxin concentration has been reported to be regulated by regulation the transcription of auxin biosynthesis genes. This is very interesting point that actually classical high temperature (28-29°C) and heat-shock (38 °C) triggered totally different responses, which is definitely worth to be further investigated in the future. Thus, as stated above in response to referee #1, we preferred to publish the data related to heat stress part after we figure out its detail mechanism and biological significance in the future. We have added the data about auxin modulates the phosphorylation of TAA1 partially via TMK4 in our manuscript, and mentioned the possible roles of this mechanism in response to environmental changes in the Discussion.

My other main concern was the lack of phenotypic differences between the wei8 complemented with the WT or the T101A mutant. The authors now provide a good explanation for the original results by showing that the T101A mutation not only abolishes phosphorylation, but also leads to a reduction in the affinity for PLP. In fact, they observe that when the wei8-3 TAA1 T101A plants are supplemented with PLP, their root meristem is significantly larger than that of the control and more similar to that of the tmk4-1 mutant. I also find these results interesting enough to be included in this manuscript.

Response: Thanks for encouraging comments from the referee. As suggested, we have added the data that the exogenous applied PLP influenced the root meristem size in *pTAA1-TAA1^{T101A};wei8-3* transgenic plant in Supplementary Fig.5e.

Overall, I believe this manuscript provides convincing lines of evidence for the regulation of TAA1 activity by phosphorylation, and that this phosphorylation is mediated by TMK4. In my opinion, these are important findings that have a significant impact on our current understanding of the regulation of auxin biosynthesis and will have important implications in shaping future research in this area.

Response: Thanks again for encouraging comments from the referee.

References:

Gourdji, S.M., Sibley, A.M. & Lobell, D.B. Global crop exposure to critical high temperatures in the reproductive period: historical trends and future projections. *Environ. Res. Lett.* **8**, 024041 (2013).

Wu, C. et al. Roles of phytohormone changes in the grain yield of rice plants exposed to heat: a review. *PeerJ* **7**, e7792 (2019).

Sharma L. et al. Auxin protects spikelet fertility and grain yield under drought and heat stresses in rice. *Environmental and Experimental Botany* **150**, 9–24 (2018).